RESEARCH COMMUNICATION

# Estimating the burden of α-thalassaemia in Thailand using a comprehensive prevalence database for Southeast Asia

Carinna Hockham[1,2]*, Supachai Ekwattanakit[3], Samir Bhatt[4], Bridget S Penman[5], Sunetra Gupta[2], Vip Viprakasit[3,6], Frédéric B Piel[7]

[1]The George Institute for Global Health, Sydney, Australia; [2]Evolutionary Ecology of Infectious Disease Group, Department of Zoology, University of Oxford, Oxford, United Kingdom; [3]Thalassaemia Centre, Faculty of Medicine, Siriraj Hospital, Mahidol University, Bangkok, Thailand; [4]Department of Infectious Disease Epidemiology, School of Public Health, Imperial College, London, United Kingdom; [5]Warwick Infectious Disease Epidemiology Research, School of Life Sciences, Warwick University, Coventry, United Kingdom; [6]Department of Paediatrics, Faculty of Medicine, Siriraj Hospital, Mahidol University, Bangkok, Thailand; [7]MRC-PHE Centre for Environment and Health, Department of Epidemiology and Biostatistics, School of Public Health, Imperial College London, London, United Kingdom

**Abstract** Severe forms of α-thalassaemia, haemoglobin H disease and haemoglobin Bart's hydrops fetalis, are an important public health concern in Southeast Asia. Yet information on the prevalence, genetic diversity and health burden of α-thalassaemia in the region remains limited. We compiled a geodatabase of α-thalassaemia prevalence and genetic diversity surveys and, using geostatistical modelling methods, generated the first continuous maps of α-thalassaemia mutations in Thailand and sub-national estimates of the number of newborns with severe forms in 2020. We also summarised the current evidence-base for α-thalassaemia prevalence and diversity for the region. We estimate that 3595 (95% credible interval 1,717–6,199) newborns will be born with severe α-thalassaemia in Thailand in 2020, which is considerably higher than previous estimates. Accurate, fine-scale epidemiological data are necessary to guide sustainable national and regional health policies for α-thalassaemia management. Our maps and newborn estimates are an important first step towards this aim.

**Editorial note:** This article has been through an editorial process in which the authors decide how to respond to the issues raised during peer review. The Reviewing Editor's assessment is that all the issues have been addressed (see decision letter).

DOI: https://doi.org/10.7554/eLife.40580.001

*For correspondence:
chockham@georgeinstitute.org.au

**Competing interests:** The authors declare that no competing interests exist.

## Introduction

α-thalassaemia is one of the commonest monogenic disorders of humans, spanning much of the malaria belt, including the Mediterranean, sub-Saharan Africa, Asia and the Pacific. It is estimated that up to 5% of the world's population carries at least one α-thalassaemia variant, with some populations (e.g. in India and Papua New Guinea) reporting gene frequencies of close to 80% (*Piel and Weatherall, 2014*). Central to its elevated frequency is the malaria protection afforded by the underlying genetic mutations, which have been favoured by natural selection in populations with historically high rates of malaria (*Flint et al., 1998*; *Premawardhena et al., 2017*; *Weatherall and Clegg, 2001a*; *May et al., 2007*). Due to recent population migrations, α-thalassaemia is now common in

other parts of the world, as illustrated by the inclusion of haemoglobin H (HbH) disease (a form of α-thalassaemia) in the newborn screening programme in California (*Vichinsky, 2013*; *Hoppe, 2009*).

Humans typically possess four copies of the α-globin gene. In an individual with α-thalassaemia, at least one of these four copies is absent or dysfunctional. The resulting deficit in α-globin affects the balance between α-globin and β- or γ-globin chains that is necessary to produce normal adult haemoglobin (HbA) and normal foetal haemoglobin (HbF), respectively (*Weatherall and Clegg, 2001b*). The severity of α-thalassaemia is inversely related to the number of functional copies of the α-globin gene. A deficit of three or more α-globin genes leads to the production of γ-globin tetramers, called Hb Bart's, in the foetus or β-globin tetramers, called HbH, in adults. Due to their very high oxygen affinity, neither tetramer is capable of transporting oxygen efficiently (*Galanello and Cao, 2011*). Furthermore, the instability of HbH leads to the production of inclusion bodies in red blood cells and a variable degree of haemolytic anaemia.

To date, 121 α-globin gene mutations have been identified (HbVar, http://globin.bx.psu.edu, accessed 07 July 2018). These include: (i) double gene deletions that remove both α-globin copies in a gene pair ($\alpha^0$-thalassaemia), (ii) single gene deletions that remove one α-globin copy ($\alpha^+$-thalassaemia), and (iii) non-deletional (ND) mutations that in some way inactivate the affected gene ($\alpha^{ND}$-thalassaemia). While deletions constitute the vast majority of these α-thalassaemia variants, non-deletional variants are typically associated with more severe phenotypes (*Chui et al., 2003*; *Fucharoen and Viprakasit, 2009*; *Lal et al., 2011*). However, even amongst non-deletional variants, considerable phenotypic variability is observed (*Singer, 2009*). Because the geographical distribution of β-thalassaemia largely overlaps with the distribution of α-thalassaemia, it is important to note that their co-inheritance often leads to a reduced imbalance between α-globin and β-globin chains, resulting in a milder thalassaemia phenotype (*Weatherall et al., 1981*; *Wainscoat et al., 1983*; *Kan and Nathan, 1970*; *Law et al., 2003*).

From a clinical perspective, α-thalassaemia is mostly a burden in Southeast Asia where $\alpha^0$-thalassaemia variants (e.g. $-^{SEA}$, $-^{THAI}$) are common and result in HbH disease when inherited with $\alpha^+$-thalassaemia (e.g. $-\alpha^{3.7}$ or $-\alpha^{4.2}$) or $\alpha^{ND}$-thalassaemia (e.g. Hb Constant Spring, or Hb CS, or Hb Paksé), or in Hb Bart's hydrops fetalis when inherited from both parents (*Weatherall et al., 2006*; *Chui, 2005*). Previously, HbH disease was considered to be relatively benign; however, recent evidence suggests a spectrum of mild to severe forms of HbH disease, with the worst affected individuals requiring lifelong transfusion (*Chui et al., 2003*; *Fucharoen and Viprakasit, 2009*; *Lal et al., 2011*). Hb Bart's hydrops fetalis, the most severe form of α-thalassaemia, associated with an absence of any functional α-globin genes, is almost always fatal *in utero* or soon after birth, although intra-uterine interventions and perinatal intensive care can lead to survival (*Songdej et al., 2017*).

In this context, there is a growing demand for a better understanding of the epidemiology of α-thalassaemia such that burden estimates can be calculated to guide public health decisions and assess the need for new pharmacological treatments. However, whilst several narrative reviews of the epidemiology of α-thalassaemia in Southeast Asian countries are available (*Chui, 2005*; *Fucharoen and Winichagoon, 1987*; *Fucharoen and Winichagoon, 2011*), a comprehensive review for the whole region has not been performed, making the current evidence-base patchy and incohesive. In addition, there appears to be a substantial amount of data that are available only in local data sources, which is not being accessed by the international community. Estimates of the number of newborns with severe forms of α-thalassaemia published by Modell and Darlison currently represent the only source of information on the epidemiology of thalassaemias and other inherited haemoglobin disorders at national and regional levels (*Modell and Darlison, 2008*). However, various inconsistencies have been identified in these α-thalassaemia estimates (*Piel and Weatherall, 2014*). Furthermore, they do not include most of the surveys conducted in the genomic era, which has allowed accurate diagnosis through DNA testing. Finally, haemoglobinopathies often present remarkably heterogeneous geographical distributions (*Piel et al., 2013a*; *Piel et al., 2013b*). As shown for other genetic conditions (e.g. sickle-cell anaemia), these variations can be captured by generating continuous allele frequency maps interpolated from population surveys using geostatistical techniques. Combined with high-resolution demographic and birth rate data, these maps allow sub-national newborn estimates to be calculated (*Piel et al., 2013a*; *Piel et al., 2013b*; *Howes et al., 2012*).

The aims of this study are therefore three-fold: i) to compile a geodatabase of published evidence for the distribution of α-thalassaemia and its common genetic variants in Southeast Asia, (ii) to

generate the first model-based continuous maps of α-thalassaemia in Thailand and calculate refined estimates of the annual number of newborns affected by severe forms of α-thalassaemia, and iii) to comprehensively evaluate and summarise the compiled evidence-base for the whole region.

## Results

### The database

Our keyword searches yielded a total of 868 unique potential sources of data on α-thalassaemia prevalence and/or genetic diversity in 10 Southeast Asian countries: Brunei Darussalam, Cambodia, Indonesia, Lao People's Democratic Republic, Malaysia, Myanmar, Philippines, Singapore, Thailand and Vietnam (*Figure 1—figure supplement 1*). A further 74 potential data sources were identified by one of the authors (SE) from local Thai journals and independently double-checked for inclusion into the study (CH). Of all these sources, 75 met our inclusion criteria and were included in the final database. Due to some sources reporting estimates for more than one population, data were available for 106 individual population samples: 58 from the online literature search and 48 from the local literature. A detailed description of the database is provided in Appendix 5.

Forty-six surveys provided data on α-thalassaemia gene frequency alone, two provided data only relating to genetic diversity and 58 provided data on both. Four surveys were reported at the national level (two from Malaysia, one from Singapore and one from Thailand), and were retained for the regional analysis. The spatial and temporal distributions of identified surveys are shown in *Figure 1—figure supplement 2*. The country for which the highest number of surveys was published in the international literature (i.e. excluding surveys obtained through local searches) was Thailand. No published surveys were identified for Brunei Darussalam or the Philippines. Within countries, surveys predominated in certain areas. For instance, in Thailand, the northern and northeastern parts of the country contained >75% of all surveys. Data for southern Thailand came exclusively from Thai journals ($n = 4$) (*Figure 1—figure supplement 2*). The total number of individuals sampled was 133,649 (the population of the region is estimated to be 647,483,729 in 2017), with a mean sample size of 1261. Further details on the final database are provided in Appendix 5.

Prevalence surveys varied considerably with regards to the α-thalassaemia alleles and/or genotypes that were tested for or reported upon; whilst some reported allele frequencies for $\alpha^0$-, $\alpha^+$- and $\alpha^{ND}$-thalassaemia, others provided data for only one or two of these. To maximise use of the available allele frequency data, whilst avoiding the incorporation of potentially biased estimates for overall α-thalassaemia allele frequency, we generated separate maps for each of the three major forms of α-thalassaemia – that is, $\alpha^0$-, $\alpha^+$- and $\alpha^{ND}$-thalassaemia (*Figure 1A,B and C*, respectively). $\alpha^0$-thalassaemia was the most extensively studied form ($n = 97$), followed by $\alpha^{ND}$-thalassaemia ($n = 49$) and then $\alpha^+$-thalassaemia ($n = 47$).

### Continuous allele frequency maps for Thailand

Data for Thailand and its neighbouring countries (Cambodia, Lao PDR, Malaysia and Myanmar) formed the evidence-base for a Bayesian geostatistical model and are presented in *Figure 2—figure supplement 1*. The total number of data points available for $\alpha^0$-, $\alpha^+$- and $\alpha^{ND}$-thalassaemia was 88, 37 and 42, respectively. The data were used to generate 1 km x 1 km maps of allele frequencies of $\alpha^0$-, $\alpha^+$- and $\alpha^{ND}$-thalassaemia in Thailand (*Figure 2*). One hundred realisations of the model were performed to generate a posterior predictive distribution (PPD) for each 1 km x 1 km pixel. The mean of the PPD is displayed, along with the 95% credible interval as a measure of model uncertainty.

The maps for $\alpha^0$- and $\alpha^+$-thalassaemia indicate clear spatial heterogeneity in allele frequencies, with ranges of 0.57–4.46% and 2.43–15.03%, respectively (*Figure 2A,B*). Heterogeneity is greatest in the north of the country. For $\alpha^0$-thalassaemia, while large parts of the northernmost provinces of Chiang Rai, Phayao and Nan have predicted allele frequencies of up to 2%, allele frequencies for the neighbouring provinces of Chiang Mai, Lampang and Phrae are often twice as high (see *Figure 2—figure supplement 2* for a reference map of Thailand provinces). The allele is also predicted at frequencies of up to 4% in the northeast of the country, along a belt across most of the north of the country and in Chonburi and Rayong provinces in central Thailand. Allele frequencies below 1% are predicted throughout the southern zone. $\alpha^+$-thalassaemia has its highest predicted allele frequencies

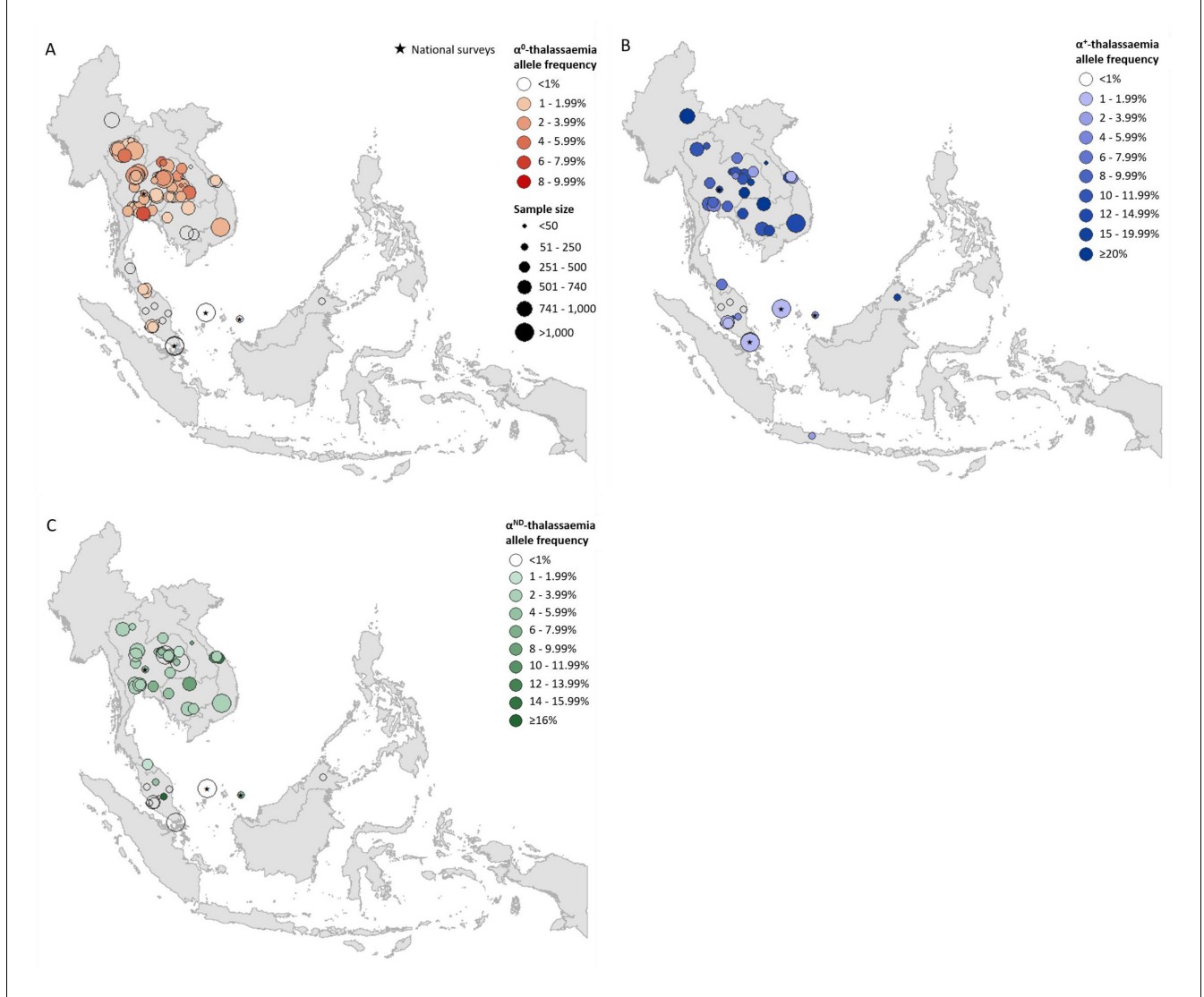

**Figure 1.** Descriptive maps of the observed allele frequencies in the database. (A) $\alpha^0$-thalassaemia, (B) $\alpha^+$-thalassaemia and (C) $\alpha^{ND}$-thalassaemia. A spatial jitter of up to $0.3^0$ latitude and longitude decimal degree coordinates was applied to allow visualisation of spatially duplicated data points. Colour intensity indicates allele frequency; circle size represents the size of the survey size. Surveys that could only be mapped at the national level are indicated by a black star.

DOI: https://doi.org/10.7554/eLife.40580.002

The following source data and figure supplements are available for figure 1:

**Source data 1.** Source data for *Figure 1A,a* map of the observed $\alpha^0$-thalassaemia allele frequencies in the database.
DOI: https://doi.org/10.7554/eLife.40580.006

**Source data 2.** Source data for *Figure 1B,a* map of the observed $\alpha^+$-thalassaemia allele frequencies in the database.
DOI: https://doi.org/10.7554/eLife.40580.007

**Source data 3.** Source data for *Figure 1C,a* map of the observed $\alpha^{ND}$-thalassaemia allele frequencies in the database.
DOI: https://doi.org/10.7554/eLife.40580.008

**Figure supplement 1.** A map of the countries included in this study.
DOI: https://doi.org/10.7554/eLife.40580.003

**Figure supplement 2.** Spatial and temporal distributions of the $\alpha$-thalassaemia surveys included in the final database.
DOI: https://doi.org/10.7554/eLife.40580.004

**Figure supplement 3.** A map of our current knowledge of the global distribution, gene frequency and genetic diversity of $\alpha$-thalassemia.
DOI: https://doi.org/10.7554/eLife.40580.005

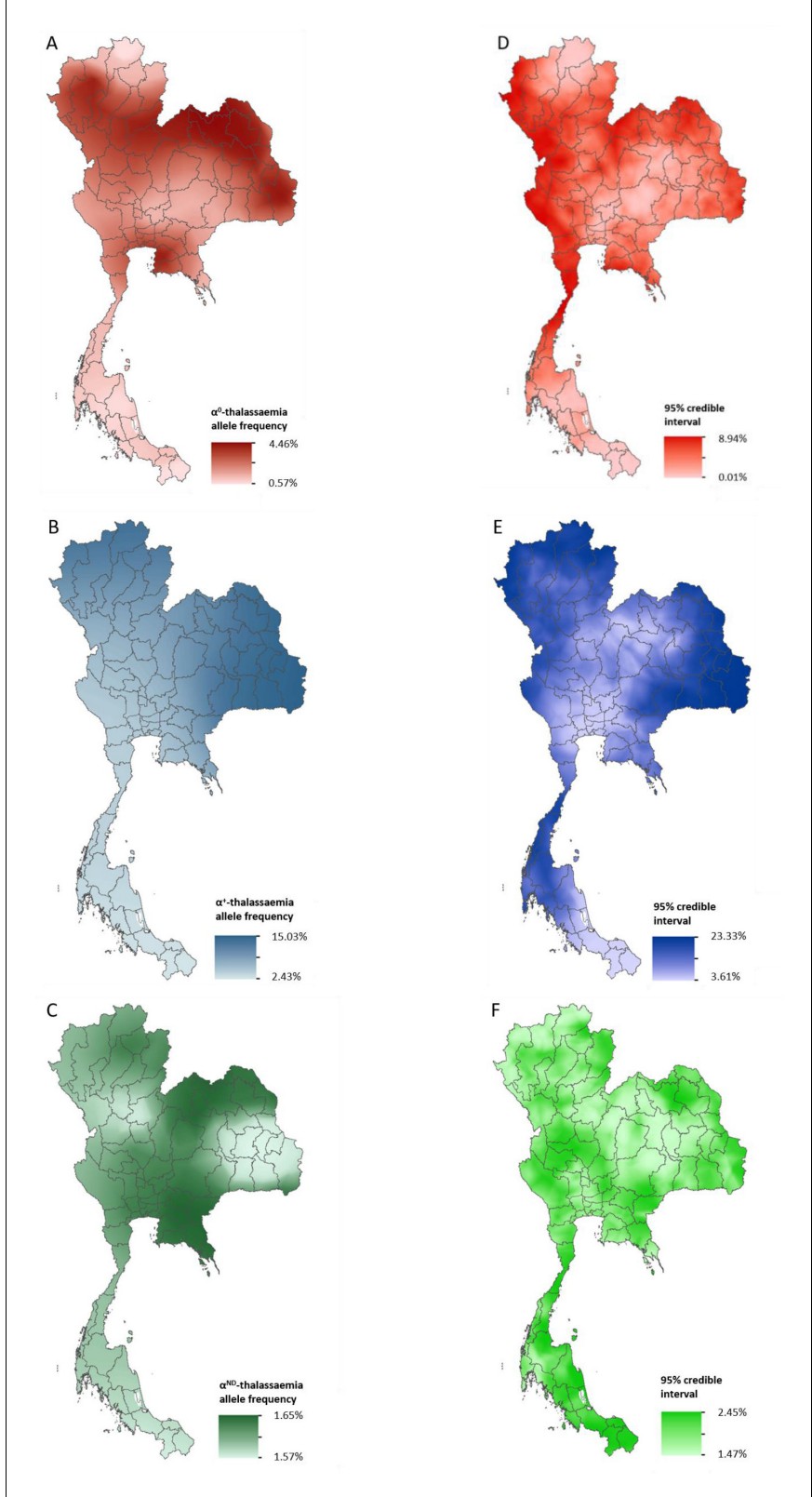

**Figure 2.** Maps of the mean of, and uncertainty in, the predicted α-thalassaemia allele frequencies in Thailand. Panels A to C display the mean of the posterior predictive distribution (PPD) of 100 realisations of the geostatistical model. Panels D to F display the 95% credible interval of the PPD. Each row corresponds to a different α-thalassaemia form: α⁰-thalassaemia (**A and D**); α⁺-thalassaemia (**B and E**) and α^{ND}-thalassaemia (**C and**

*Figure 2 continued on next page*

*Figure 2 continued*

F). *Figure 2—figure supplement 1* shows the observed data used to construct the models and *Figure 2—figure supplement 2* displays the province names for reference.
DOI: https://doi.org/10.7554/eLife.40580.009
The following figure supplements are available for figure 2:

**Figure supplement 1.** Maps of the observed allele frequencies used to construct the models and generate the predicted continuous allele frequency maps for Thailand in *Figure 2*.
DOI: https://doi.org/10.7554/eLife.40580.010
**Figure supplement 2.** A reference map of Thailand provinces.
DOI: https://doi.org/10.7554/eLife.40580.011

across the whole of the north and northeastern zones. Predicted allele frequencies of $\alpha^{ND}$-thalassaemia range between 1.57% and 1.65% only.

Model uncertainty is greatest in areas where data are scarce (e.g. southern Thailand and along the border with Myanmar) or where there is heterogeneity in the available data (e.g. in Chiang Mai and the surrounding area). Overall, uncertainties are higher for $\alpha^+$-thalassaemia than for $\alpha^0$-thalssaemia or $\alpha^{ND}$-thalassaemia, which is partly due to the wider range of observed frequencies for this form. For $\alpha^0$-thalassaemia, the highest level of uncertainty is 9% and is found in Chumphon and Ranong provinces in southern Thailand and Kanchanaburi and Tak in the westernmost part of the country. For $\alpha^+$-thalassaemia, the highest uncertainty (up to 23%) is observed in the northeastern zone and in the north. Uncertainty for $\alpha^{ND}$-thalassaemia is patchy and ranged from 1.5% in central and northern Thailand to 2.5% in southern and northeastern Thailand. The results of the 10-fold cross-validation procedure reveal an average mean absolute error of the predictions of 0.93%, 4.10% and 2.30%, for $\alpha^0$-, $\alpha^+$- and $\alpha^{ND}$-thalassaemia, respectively. The average correlation between the observed and predicted values is 0.74 (0.62–0.83), 0.71 (0.49–0.85) and 0.47 (0.17–0.69), respectively.

## Estimates of number of affected newborns in Thailand

Estimates of the number of newborns born with a severe form of $\alpha$-thalassaemia (i.e. Hb Bart's hydrops fetalis and HbH disease) in Thailand in 2020 were generated by pairing our allele frequency predictions to high-resolution demographic data for the country. We estimate that the number of Hb Bart's hydrops fetalis births in the country will be 423 (CI: 184–761) in 2020. The number of new cases of HbH disease is estimated to be 3,172, including 2674 (CI: 1,296–4,491) deletional and 498 (CI: 237–947) non-deletional cases. The highest absolute burden of hydrops fetalis is predicted in Bangkok City (57 [CI: 13–151]) (*Figure 2—figure supplement 2*), with its high population density, and Udon Thani (23 [CI: 6–66]) in the northeastern zone, where some of the highest $\alpha^0$-thalassaemia allele frequencies are predicted. Other provinces with a comparatively high burden include: Chiang Mai in the north of the country; Khon Kaen, Sakon Nakhon and Ubon Ratchathani in the northeast; and Chon Buri, Samut Prakan and Nonthaburi close to Bangkok City. The estimated number of hydrops fetalis births in these provinces range between 10 and 19. For HbH disease, the highest burden is predicted in northeast Thailand for both the deletional and non-deletional forms. Bangkok City is predicted to have the highest burden of HbH disease (301 [CI: 94–639] for deletional HbH disease and 68 [CI: 25–148] for non-deletional HbH disease).

To directly compare estimates generated using our methodology with those previously published by Modell and Darlison, we also calculated estimates using population and birth rate data for 2003 (Appendix 3). Modell and Darlison estimated 1017 and 2515 births to be affected by Hb Bart's hydrops fetalis and HbH disease, respectively, in 2003. Using population data from the same year paired with our model-based maps, and assuming no consanguinity, we estimate 709 and 5469 newborns to be born with Hb Bart's hydrops fetalis and HbH disease in the country. As Modell and Darlison included a population coefficient of consanguinity (*F*) in their calculations, we examined the effect that this would have on our estimates. We found that they do not change considerably (951 and 5,409), when a value of *F* of 0.1, a high value for the region (www.consang.net), is incorporated. Our estimates are therefore consistent with those by Modell and Darlison for Hb Bart's hydrops fetalis. However, they suggest that the burden of HbH disease in Thailand may have previously been underestimated. Moreover, whilst Modell and Darlison did not estimate the burden of non-

deletional forms of HbH disease, our estimates suggest that 15% of the 5469 neonatal cases were of non-deletional types, which are usually associated with more severe phenotypes.

## Overall distribution of α-thalassaemia across Southeast Asia

In our database for all of Southeast Asia, the number of surveys that tested for all three forms of α-thalassaemia was 40. Amongst these, the overall α-thalassaemia gene frequency ranged from 0% in populations from peninsular Malaysia to 35.4% in Preah Vihar, Cambodia (*Munkongdee et al., 2016*). A higher allele frequency of 49% was also reported in the So ethnic group from Khammouane Province in Lao PDR, although the sample size for this study was small (*n* = 50) (*Sengchanh et al., 2005*). *Appendix 5—table 2* shows the range of allele frequencies observed for the different α-thalassaemia forms ($\alpha^0$-, $\alpha^+$- and $\alpha^{ND}$-thalassaemia) in each country.

For $\alpha^0$-thalassaemia, the highest allele frequencies were observed in Thailand (*Figure 1A*, *Figure 1—source data 1*) In Lao PDR, surveys along the Lao PDR-Thailand border near Vientiane reported allele frequencies between 4.03% and 7.28%, whilst the survey among the aforementioned So ethnic group reported an absence of the $\alpha^0$-thalassaemia allele. The highest reported allele frequencies in Cambodia and Vietnam were 1.10% and 2.66%, respectively, with the majority of studies reporting even lower frequencies. However, data were sparse in the two countries (*n* = 4 in each). Allele frequencies of up to 1.53% were observed in southern Thailand, whilst the few surveys carried out in Myanmar (*n* = 1), Malaysia (*n* = 11) and Singapore (*n* = 2) reported allele frequencies of around 1%. In Malaysia, the highest allele frequency of $\alpha^0$-thalassaemia was 1.92% from a study carried out in newborns in Kuala Lumpur, the capital city (*Alauddin et al., 2017*).

$\alpha^+$-thalassaemia, the most prevalent form, reached allele frequencies of 26% in Cambodia (*Figure 1B*, *Figure 1—source data 2*) (*Munkongdee et al., 2016*). The surveys revealed a clear north-to-south decline in the distribution of $\alpha^+$-thalassaemia across the region, with a single high allele frequency estimate of 16.8% observed in Sabah in Malaysian Borneo (*Tan et al., 2010*). High allele frequencies ($\geq$10%) were observed in all four surveys in Cambodia. In Vietnam the reported $\alpha^+$-thalassaemia allele frequency ranged from 1.59 to 14.4%.

The observed allele frequency of $\alpha^{ND}$-thalassaemia ranged between 0% in various locations across Malaysia and 16.25% in central Peninsular Malaysia (*Figure 1C*, *Figure 1—source data 3*). Within Thailand, the highest reported allele frequencies of around 7% were observed in Khon Kaen in the northeast and Chachoengsao in central Thailand. In Cambodia and Vietnam, $\alpha^{ND}$-thalassaemia allele frequencies of up to 8% and 14.3% were reported, respectively, in surveys in which $\alpha^0$-thalassaemia was found to be absent, whilst in other parts of these countries, the two forms were found to co-exist at similar allele frequencies (e.g. around 2.5% in Binh Phuoc and Khanh Hoa provinces in Vietnam).

## Genetic diversity of α-thalassaemia across Southeast Asia

Maps of the genetic diversity of α-thalassaemia across Southeast Asia are shown in *Figures 3–6*. *Figure 3* (*Figure 3—source data 1*) displays surveys that included all three α-thalassaemia forms ($\alpha^0$-, $\alpha^+$- and $\alpha^{ND}$-thalassaemia), allowing relative proportions of each of the forms to be calculated without giving specific variant details. *Figures 4–6* (*Figure 4—source data 1*, *Figure 5—source data 1*, *Figure 6—source data 1*) display surveys that provided information on the *frequencies* of specific α-thalassaemia variants (e.g. $-^{SEA}$, $-\alpha^{3.7}$, etc.). For these, the variants that were tested for differ between surveys. Some surveys are included in both *Figure 3* and *Figures 4–6*. For the latter figures, the region has been divided to improve visualisation of the data.

$\alpha^+$-thalassaemia most consistently constituted the highest proportion of α-thalassaemia, although there were some surveys in which $\alpha^{ND}$-thalassaemia was the predominant form (e.g. central Vietnam and in parts of Malaysia). In *Figure 3*, areas where the observed relative proportion of $\alpha^0$-thalassaemia was greatest include: Chiang Mai and Phayao provinces in north Thailand, Kalasin in northeast Thailand, Vientiane in Lao PDR, Kuala Lumpur and Selangor in Malaysia, Singapore and Jakarta in Indonesia. The $\alpha^0$-thalassaemia allele was absent in the survey from Malaysian Borneo as well as in central Vietnam and central Lao PDR. In certain areas, $\alpha^0$- and $\alpha^{ND}$-thalassaemia together accounted for the majority of α-thalassaemia (e.g. ~75% in Kalasin in Thailand, ~60% in Kuala Lumpur and Jakarta and ~53% in Khon Kaen and Chachoensao in Thailand and Vientiane in Lao PDR). Some of

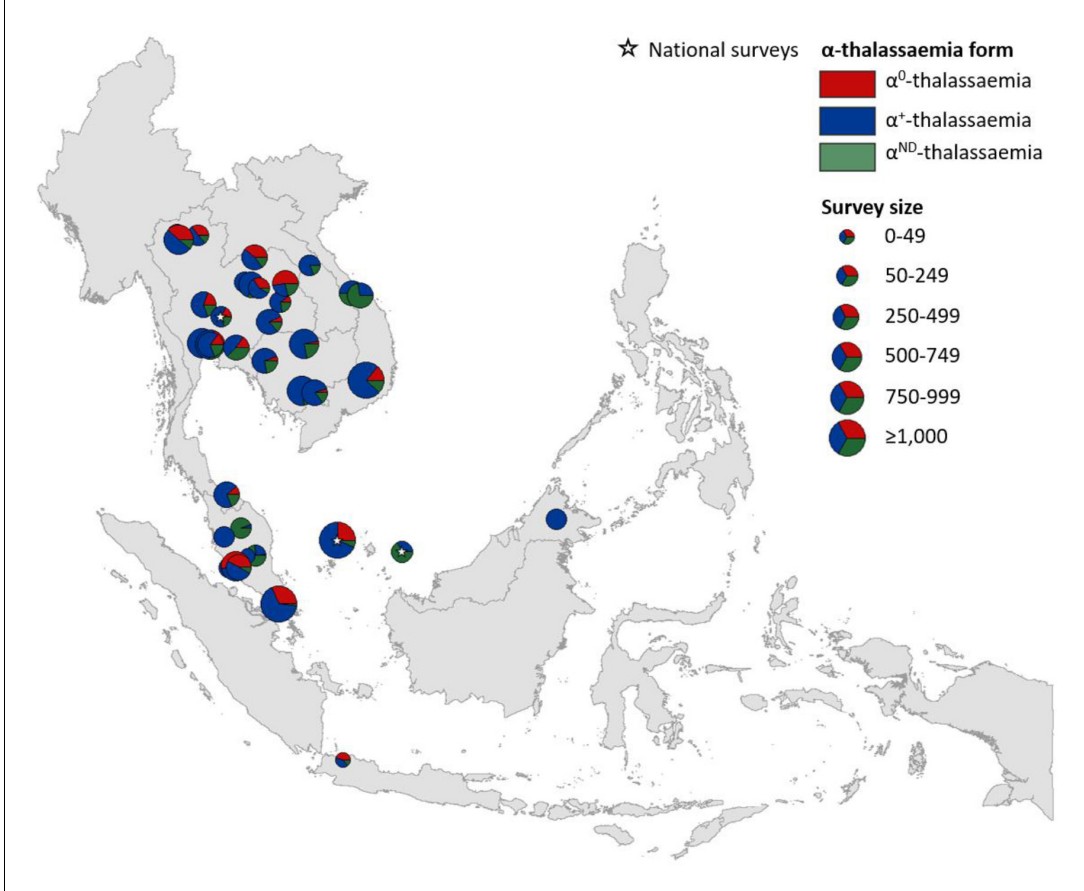

**Figure 3.** Map showing the proportions of $\alpha^0$-, $\alpha^+$- and $\alpha^{ND}$-thalassaemia in Southeast Asia. Three surveys were mapped at the national level (indicated by a white star). The size of the pie charts reflects survey size.

DOI: https://doi.org/10.7554/eLife.40580.014

The following source data is available for figure 3:

**Source data 1.** Source data for *Figure 3*, a map showing the proportions of $\alpha 0$-, $\alpha +$- and $\alpha ND$-thalassaemia in Southeast Asia.

DOI: https://doi.org/10.7554/eLife.40580.015

these areas also correspond to where the highest allele frequencies of these alleles are found, for example, northeast Thailand and the Thailand-Lao PDR border.

Among surveys that tested for specific $\alpha$-thalassaemia variants, the most commonly tested variant throughout the region was $-^{SEA}$ ($n = 44$), followed by $-\alpha^{3.7}$ ($n = 36$) (*Figures 4–6*). Most of the studies in Thailand only tested for a subset of the mutations considered in this study; only one survey in Bangkok City tested for the whole suite. More than in other countries, surveys in Thailand tested specifically for $\alpha^0$- or $\alpha^{ND}$-thalassaemia mutations ($n = 7$ and 9, respectively).

Throughout the region, $-^{SEA}$ was the dominant $\alpha^0$-thalassaemia mutation, and in the majority of surveys $-\alpha^{3.7}$ was the dominant $\alpha^+$-thalassaemia and Hb CS the dominant $\alpha^{ND}$-thalassaemia mutation. The only exceptions were in Java in Indonesia, where $-\alpha^{3.7}$ and $-\alpha^{4.2}$ were found at similar frequencies and in Kelantan in Malaysia, where Hb Adana was the only $\alpha^{ND}$-thalassaemia mutation identified. The $-(\alpha)^{20.5}$ and $-^{MED}$ mutations were not detected in any of the surveys, whilst the $-^{FIL}$ mutation was found in 2 of the 16 surveys in which it was tested for and $-^{THAI}$ in 9 of the 31 surveys in which it was included. Consistent with *Figure 3*, $\alpha^0$-thalassaemia variants contributed minimally to $\alpha$-thalassaemia mutations in Myanmar, Cambodia and Vietnam but were found at higher frequencies in surveys along the Thailand-Lao PDR border. In Vietnam, Malaysia, Indonesia and Singapore, the frequency of $\alpha^0$-thalassaemia varied considerably, with it being absent in some areas and a predominant form in others. This is also true for $\alpha^{ND}$-thalassaemia.

## Discussion

α-thalassaemia is a neglected public health problem whose burden has, to date, been largely over-looked, but for which morbidity is expected to increase in the coming decades as a result of the epidemiological transition, whereby acute infectious disease is replaced by chronic disease as the predominant cause of morbidity and mortality (*Piel and Weatherall, 2014*; *Weatherall, 2011*). Moreover, country reports (e.g. from Malaysia) indicate a shift in the age distribution of thalassaemia patients towards older ages (*Ibrahim, 2012*). As the burden increases, there will be greater demand for resources, including healthcare facilities and staff, genetic counselling and drugs, to treat and manage affected patients. This is particularly true for countries in Southeast Asia as well as the Mediterranean, where severe forms of α-thalassaemia (i.e. $\alpha^0$-thalassaemia) are found.

### Comparison with existing maps and population estimates

The model-based maps for Thailand presented here are, to our knowledge, the first spatially continuous maps of the distribution of α-thalassaemia in any country. Our newborn estimates represent the first evidence-based estimates of specific forms of α-thalassaemia disease amongst newborns since 2003 (although the study in which they were reported was published in 2008) (*Modell and Darlison, 2008*) and the first estimates at sub-national level. Importantly, whilst there are currently no estimates of the number of stillbirths that will occur in Thailand in 2020, our estimate of the number of Hb Bart's hydrops fetalis births represents more than 10% of the 3697 stillbirths estimated for 2015 (*Blencowe et al., 2016*).

Comparisons between previous newborn estimates and those generated in this study using our updated database and 2003 demographic data revealed an almost two-fold difference for deletional HbH disease (2515 compared to 4694 in the present study). Reasons for such discrepancies most likely relate to: (i) differences in the inclusion criteria used in the generation of our map and therefore our calculation of newborn estimates, (ii) the quantity of survey data used, and iii) the statistical methods employed. For instance, spatial specificity was not a consideration in the study by Modell and Darlison, who used a single allele frequency estimate extrapolated to the whole country. As such, the newborn calculations in the present study represent a methodological advance over previous efforts to assess the burden of α-thalassaemia. We related fine-scale allele frequency data to birth count data of equally high resolution, allowing location-specific estimates to be generated that could be aggregated to province level. Moreover, the use of model-based maps in our calculations enabled the measurement of uncertainty in our predictions. Finally, by including allele frequency data on $\alpha^{ND}$-thalassaemia, we were able to estimate the burden of the more severe non-deletional HbH disease.

Our newborn estimates for 2020 are considerably lower than those for 2003. This reduction is due to the lower number of births in Thailand in 2020 as a result of a decreasing birth rate and population size (*World population prospects, 2017*). It would be interesting to quantify how improvements in the prevention of thalassaemias will affect these estimates in the future.

Our descriptive maps represent the first detailed cartographic representations of α-thalassaemia allele frequency estimates in Southeast Asia, which take into account the specific geographical location of the surveys in which they were observed. Until now, available maps (e.g. *Figure 1—figure supplement 3*) provided only a crude overview of overall α-thalassaemia gene frequency, without any distinction between different α-thalassaemia forms, and extrapolated to the entire region, thereby masking sub-national and even international, variation in allele frequencies (*Piel and Weatherall, 2014*).

The maps are broadly consistent with early narrative reviews of the gene frequency of α-thalassaemia in the region (*Fucharoen and Winichagoon, 1987*), showing a clear north-to-south trend of decreasing allele frequencies of $\alpha^0$- and $\alpha^+$-thalassaemia and a patchier distribution of $\alpha^{ND}$-thalassaemia. However, our maps also demonstrate a severe lack of data on the allele frequency of α-thalassaemia across large parts of Southeast Asia, including in Myanmar, northern Lao PDR, northern Vietnam, Indonesia, Philippines and Brunei. This impedes our ability to assess the fine-scale burden of α-thalassaemia, making efficient public health planning for its control difficult, and limits our ability to track progress in the prevention and management of the disorder.

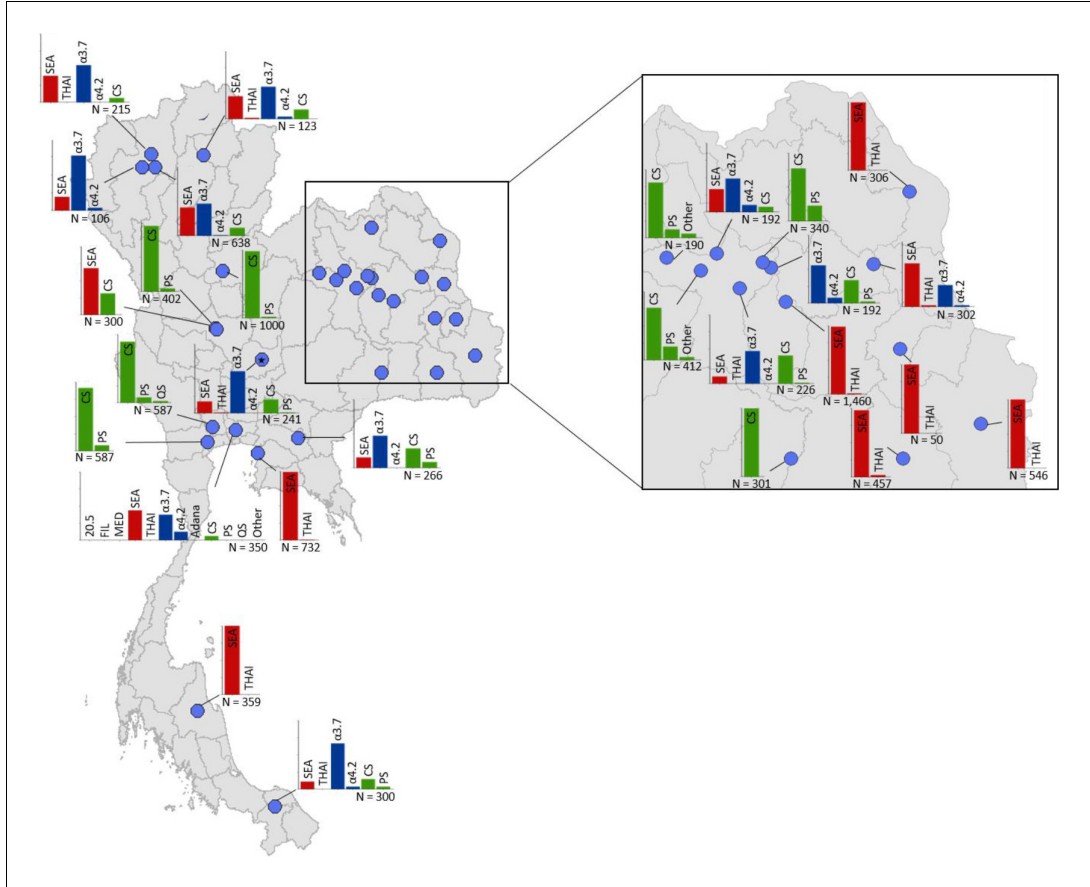

**Figure 4.** Map showing the allele frequencies of specific α-thalassaemia variants in Thailand. Given the high number of surveys in northeast Thailand, this region has been magnified. The y-axis scale is the same across all bar charts, ranging from 0 to 1. The variants that were tested for in each survey are indicated above each bar. $\alpha^0$-thalassaemia mutations are shown in red, $\alpha^+$-thalassaemia mutations in blue and $\alpha^{ND}$-thalassaemia mutations in green. Empty spaces along the x-axis indicate an absence of the corresponding mutation in the survey sample. The sample size of the survey is given under each plot. Bar charts are connected to their spatial location by a black line.

DOI: https://doi.org/10.7554/eLife.40580.016

The following source data is available for figure 4:

**Source data 1.** Source data for *Figure 4*, a map showing the proportions of specific α-thalassaemia variants in Thailand.

DOI: https://doi.org/10.7554/eLife.40580.017

## Patterns of genetic variation and their public health implications

The pattern of genetic diversity observed in this study indicates variable distributions of mild and severe α-thalassaemia forms. Reasons for this are unclear. However, high variant heterogeneity has been observed for other genetic disorders (e.g. G6PD deficiency) in Southeast Asia, (*Howes et al., 2013*) which might suggest a similar underlying cause. In their global study, Howes *et al.* noted that G6PD variants were most diverse in East Asia and the West Pacific, where *P. falciparum* parasites show strong population structure with lower genetic relatedness between populations in the region. Indeed, *P. falciparum* has been shown to display genetically structured populations within Thailand alone. (*Pumpaibool et al., 2009*) It is possible that the evolutionary dynamics between *P. falciparum* and haemoglobin variants, including α-thalassaemia, are more complex than we currently appreciate.

The observed spatial distributions of the different α-thalassaemia forms and variants has important implications for the design of newborn screening programmes with regards to the preferred

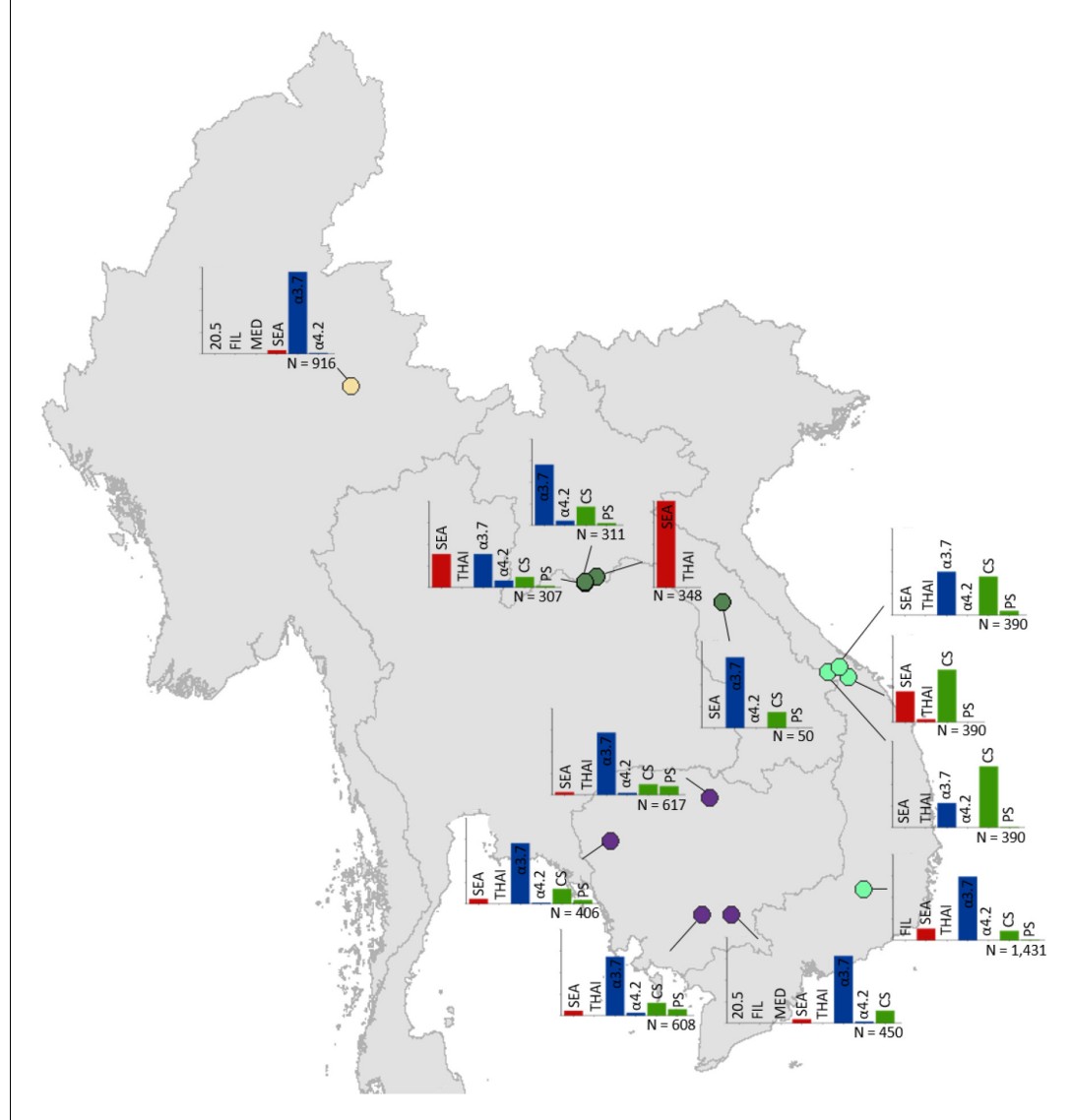

**Figure 5.** Map showing the allele frequencies of specific α-thalassaemia variants in Myanmar, Lao PDR, Cambodia and Vietnam. The y-axis scale is the same across all bar charts, ranging from 0 to 1. The variants that were tested for in each survey are indicated above each bar. $\alpha^0$-thalassaemia mutations are shown in red, $\alpha^+$-thalassaemia mutations in blue and $\alpha^{ND}$-thalassaemia mutations in green. Empty spaces along the x-axis indicate an absence of the corresponding mutation in the survey sample. The sample size of the survey is given under each plot. Bar charts are connected to their spatial location by a black line. Data points are coloured by country, using the same colour scale as that in *Figure 1—figure supplement 1*.

DOI: https://doi.org/10.7554/eLife.40580.012

The following source data is available for figure 5:

**Source data 1.** Source data for *Figure 5*, a map showing the proportions of specific α-thalassaemia variants in Cambodia, Lao PDR, Myanmar and Vietnam.
DOI: https://doi.org/10.7554/eLife.40580.013

diagnostic algorithm and allocation of treatment and management service provision. Areas with the highest proportions of co-occurring severe α-thalassaemia forms (i.e. $\alpha^0$-thalassaemia and $\alpha^{ND}$-thalassaemia) may experience a higher prevalence of the severe non-deletional form of HbH disease. Furthermore, the predominance of Hb CS in surveys from Malaysia and Vietnam suggests that the health burden of α-thalassaemia in these areas may be greater than previously thought. Hb CS is a mutation at the termination codon of the α2-globin gene, which, in a normal individual, accounts for three-quarters of overall α-globin production (*Liebhaber and Kan, 1981*; *Orkin et al., 1981*). As a

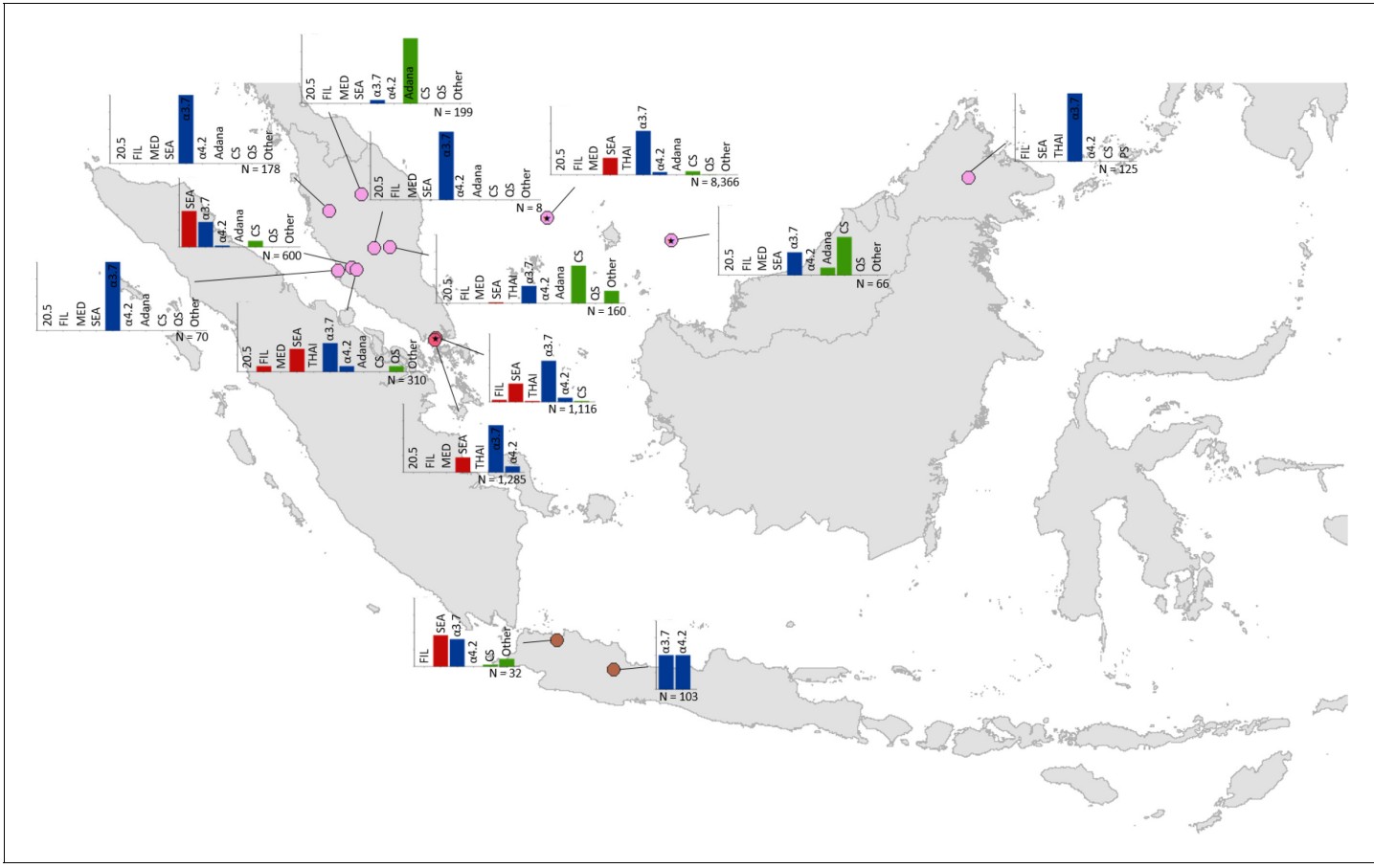

**Figure 6.** Map showing the allele frequencies of specific α-thalassaemia variants in Malaysia, Singapore and Indonesia. The y-axis scale is the same across all bar charts, ranging from 0 to 1. The variants that were tested for in each survey are indicated above each bar. $\alpha^0$-thalassaemia mutations are shown in red, $\alpha^+$-thalassaemia mutations in blue and $\alpha^{ND}$-thalassaemia mutations in green. Empty spaces along the x-axis indicate an absence of the corresponding mutation in the survey sample. The sample size of the survey is given under each plot. Bar charts are connected to their spatial location by a black line. Data points are coloured by country, using the same colour scale as that in *Figure 1—figure supplement 1*.

DOI: https://doi.org/10.7554/eLife.40580.018

The following source data is available for figure 6:

**Source data 1.** Source data for *Figure 6*, a map showing the proportions of specific α-thalassaemia variants in Indonesia, Malaysia and Singapore.

DOI: https://doi.org/10.7554/eLife.40580.019

result, α2-globin gene mutations, such as Hb CS, tend to cause a more severe phenotype (*Chui, 2005*).

## Model strengths and limitations

The reliability of the model-based maps is intrinsically linked to the quality, quantity and spatial coverage of the data upon which the models are based. We were unable to generate continuous maps for the whole of the Southeast Asian region as data were sparse in large areas. Whilst we are aware that unpublished surveys are likely to be available for most countries of the region, obtaining local data for all of the countries was beyond the scope of this study. Nevertheless, we have demonstrated that substantial additional survey data can be identified in locally published sources and, as a result, highlighted the enormous value of future collaborations to collate local data in other regions.

For Thailand, limitations relating to data sparsity, uneven survey distributions and heterogeneity in allele frequency can be quantified in the presented uncertainty intervals. Areas where there is little data or where observed allele frequencies are highly heterogeneous within a small geographical area will have more uncertain predictions, whilst a large amount of data for which there is little heterogeneity will lead to more precise predictions. We identified a lack of data in the southern part of

Thailand, which is reflected in larger uncertainty estimates. Other predictions with high associated uncertainty include those along the Thailand-Myanmar border, particularly the southern tail of Myanmar, where no α-thalassaemia prevalence surveys are found. This highlights the arbitrary nature of country borders in mapping studies.

Spatial smoothing is an important component of most geostatistical models. For the modelling approach used here, the range function (i.e. the extent of spatial autocorrelation) is defined by a parameter within the SPDE framework and takes a prior distribution. The smoothing in the approximate posterior therefore balances over- and under-fitting and is necessary to ensure that the model predicts adequately without fitting the idiosyncrasies of the data. As a result, the model does not predict allele frequencies that fully reflect heterogeneity between nearby surveys. Although extensive variation in allele frequencies between different ethnic groups in similar geographic locations has been observed in Thailand (*Kulaphisit et al., 2017*) and other countries (e.g. Sri Lanka), this could not be reflected in our predicted allele frequencies. For example, allele frequencies of around 3.65% for the Hb CS mutation have been reported in the Khmer ethnic group in Surin and Buriram provinces, whilst our model predicts maximum allele frequencies of 1.65% here. This smoothing process can similarly explain why the highest observed $\alpha^{ND}$-thalassaemia frequency of 7% in Khon Kaen was not reproduced in the predicted maps. In fact, our model-based predictions for $\alpha^{ND}$-thalassaemia are remarkably homogenous and the average correlation between the observed and predicted frequencies is low (0.47). This is because the close-range heterogeneity in the observed data, coupled with the absence of a long-range trend in frequency (as is observed for $\alpha^{0}$- and $\alpha^{+}$-thalassaemia), makes it difficult for the model to discern a signal.

It is likely that other factors influence the allele frequencies of the different α-thalassaemia forms, but have not been considered in this mapping study, including ethnicity, consanguinity, historic rates of malaria (both *Plasmodium falciparum* and *P. vivax*) (*Douglas et al., 2012*) and population migration patterns. Furthermore, there is bound to be uncertainty in the geolocation of some of the surveys included in the study due to the lack of details published or available. This uncertainty could not be accounted for. Finally, whilst the inclusion criterion of molecular methods should help to improve the reliability of allele frequency estimates, they are not 100% sensitive (*Old and Henderson, 2010*) and do not cover all possible α-thalassaemia mutations, which may lead to some error in the reported allele frequencies.

Whilst we have calculated the burden of α-thalassaemia in terms of the number newborns born with severe forms in 2020, there are other aspects of the disease burden that would be worth considering pending the availability of more data, for example, milder-forms and their coinheritance with β-thalassaemia, DALY losses from α-thalassaemia, maternal complications (some of which can be life-threatening) (*Chui, 2005*; *Ratanasiri et al., 2009*), psychological effects and, in the case of HbH disease, survival data allowing the calculation of all-age population estimates. Furthermore, the estimates presented here do not include compound disorders, such as EA Bart's and EF Bart's diseases (HbH disease with heterozygous and homozygous forms of $\beta^{E}$, another clinically important structural β-globin variant, respectively), and therefore remain underestimates of the overall burden of α-thalassaemia disorders in Thailand (*Galanello, 2013*). Finally, the visualisation of our burden estimates are subject to the modifiable area unit problem, whereby the presentation of estimates at the province level likely masks pockets of high burden (*Wong, 2009*).

## Future prospects and conclusions

The allele frequency, distribution and genetic variant profile of α-globin forms is only a part of their epidemiological complexity. An improved understanding of the natural history of α-thalassaemia and the factors that modify its clinical outcome will be imperative for establishing better estimates of its burden. This is particularly pertinent in the Southeast Asian region, where the disorder co-exists with β-thalassaemia, including the commonest haemoglobin variant, Hb E. Many studies have shown a positive epistatic interaction between α- and β-thalassaemia, whereby their co-inheritance results in the amelioration of the associated blood disorder (*Fucharoen and Weatherall, 2012*; *Viprakasit et al., 2004*).

A detailed assessment of current knowledge on the allele frequency of α-thalassaemia and the magnitude of its health burden is needed to develop suitable prevention and control programmes. This study provides a detailed overview of the existing data on the gene frequency and genetic diversity of α-thalassaemia in Southeast Asia. We show that our knowledge of the accurate allele

frequency and distribution of this highly complex disease remains somewhat limited. Because of the remarkable geographic heterogeneities in the gene frequency of α-thalassemia, interventions have to be tailored to the specific characteristics of the local population (e.g., prevalence of the disorder in the population, ethnic makeup, and consanguinity) and the local health care system As the epidemiological transition in these countries continues (*Weatherall, 2011*; *Bundhamcharoen et al., 2011*; *Dhillon et al., 2012*), it will become increasingly important to regularly update regional and national maps of α-thalassaemia gene frequency and newborn estimates such that health and demographic changes can be properly quantified (*Piel and Weatherall, 2014*). Our findings provide a baseline for such endeavours.

## Materials and methods

### Compiling a geodatabase of α-thalassaemia allele frequency and genetic diversity

A comprehensive search of three major online bibliographic databases (PubMed, ISI Web of Knowledge and Scopus) was performed to identify published surveys of α-thalassaemia prevalence and/or genetic diversity in Southeast Asia (*Figure 7*). The 10 member states of the Association of Southeast Asian Nations (ASEAN) were used to define the region under study and include: Brunei Darussalam, Cambodia, Indonesia, Lao PDR, Malaysia, Myanmar, Philippines, Singapore, Thailand and Vietnam (*Figure 1—figure supplement 1*). In addition, for Thailand, articles published in national journals (in Thai) – not included in international bibliographic databases – were manually searched for local surveys. Consistent and pre-defined sets of inclusion criteria for prevalence/allele frequency data and genetic diversity data, outlined in the Appendix 1, were used to identify relevant surveys. Data

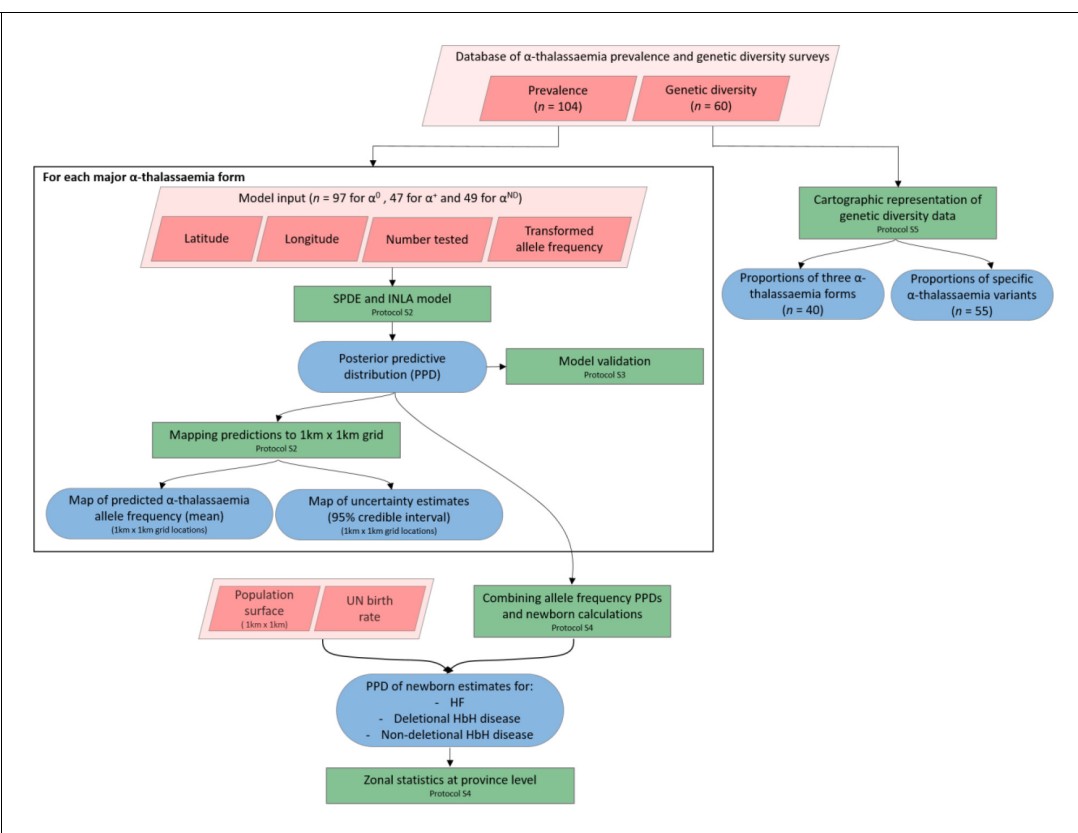

**Figure 7.** A schematic overview of the methodology used in this study and a breakdown of the data types analysed. Pink diamonds indicate the database and input data; green boxes denote model processes and data visualisation steps; blue rods represent study outputs. .
DOI: https://doi.org/10.7554/eLife.40580.020

extracted from Thai journals were independently validated against the inclusion criteria by two of the authors (SE and CH).

## Modelling continuous maps of α-thalassaemia allele frequency in Thailand

We employed a Bayesian geostatistical framework to model the allele frequencies of $\alpha^0$-, $\alpha^+$- and $\alpha^{ND}$-thalassaemia, respectively, in Thailand, where a substantially higher number of surveys were identified. We included data from Thailand and its neighbouring countries (Myanmar, Lao PDR, Cambodia and Malaysia) in order to preclude the possibility of a border effect. Three surveys that were reported only at the national level (one in Thailand and two in Malaysia) were excluded for this part of the analysis. Only geographical location was included as a predictor of α-thalassaemia allele frequency (*Figure 7*).

For each of the three main forms of α-thalassaemia, a model was fitted using a Bayesian Stochastic Partial Differential Equation (SPDE) approach with Integrated Nested Laplace Approximation (INLA) algorithms developed by Rue *et al*. (*Rue et al., 2009*), available in an R-package (www.r-inla.org). The observed allele frequency data were transformed through an empirical logit to facilitate approximation by the Gaussian likelihood. The fitted model was then used to generate predictions at a resolution of 1 km x 1 km for α-thalassaemia allele frequencies for all unsampled locations in Thailand. Uncertainty estimates, measured as the 95% credible interval, for the predictions were calculated using 100 conditionally simulated realisations of the model to generate a posterior predictive distribution (PPD) for each 1 km x 1 km pixel. Full details of the modelling process and model validation procedures, which involved a 10-fold cross validation, are provided in Appendix 2.

## Refining estimates of the annual number of neonates affected by severe disease forms

To generate estimates of the annual number of newborns affected by Hb Bart's hydrops fetalis syndrome (–/–) and deletional and non-deletional HbH disease (-α/– and $\alpha\alpha^{ND}$/–, respectively) in Thailand in 2020, we paired the predicted allele frequency maps generated using our Bayesian geostatistical framework with high-resolution birth count data. First, we combined the three allele frequency maps to estimate the frequency of each genotype in each pixel, assuming Hardy-Weinberg proportions for a four-allele system (*Equation 1* and *Table 1*) (*Hardy, 1908*; *Weinberg, 1908*).

$$p^2 + 2pq + 2pr + q^2 + 2qr + 2qs + r^2 + 2rs + s^2 = 1 \tag{1}$$

where, p is the allele frequency of α $\alpha^0$-thalassaemia (–), q is the allele frequency of $\alpha^+$-thalassaemia (-α), r is the allele frequency of $\alpha^{ND}$-thalassaemia ($\alpha\alpha^{ND}$) and s is the allele frequency of the wild-type α-globin haplotype (αα).

To calculate birth counts, the 2015–2020 crude birth rate for Thailand was downloaded from the 2017 United Nations (UN) world population prospects (*World population prospects, 2017*) and multiplied with a high-resolution predicted 2020 population surface, adjusted to UN population estimates, obtained from the WorldPop project (www.worldpop.org.uk, last accessed 23 January 2018) (*Tatem, 2017*). The predicted genotype frequencies were then paired with the birth count data over 100 conditionally simulated realisations of the geostatistical model and areal estimates at province

**Table 1.** A breakdown of the genotypes for the three clinically important forms of α-thalassaemia – Hb Bart's hydrops fetalis, deletional HbH disease and non-deletional HbH disease – and the Hardy-Weinberg equilibrium (HWE) proportions used for their calculation.

To compare our model output with previous newborn estimates for Hb Bart's hydrops fetalis and deletional HbH disease, we paired our allele frequency maps with 2003 demographic and birth data and included a measure of consanguinity in our calculations.

| Genotype | Disorder | HWE proportions | Inclusion of population coefficient of consanguinity (F) |
|---|---|---|---|
| –/– | Hb Bart's hydrops fetalis | $p^2$ | $p^2 + Fp(1\ p)$ |
| -α/– | Deletional HbH disease | $2pq$ | $2pq(1\ F)$ |
| $\alpha\alpha^{ND}$/– | Non-deletional HbH disease | $2pr$ | $2pr(1\ F)$ |

DOI: https://doi.org/10.7554/eLife.40580.021

level calculated, together with 95% credible intervals; their calculation is described in Appendix 3. We also applied our maps to 2003 demographic data, and incorporated consanguinity into our calculations (*Table 1*), (*Vieira et al., 2013*) in order to more directly compare estimates generated using our method with previous estimates (*Modell and Darlison, 2008*). We used the online global database of consanguinity estimates (www.consang.net) to identify an upper limit for the coefficient of consanguinity for Thailand ($F = 0.1$) (*Bittles and Black, 2015*). However, due to important variations of this coefficient between ethnic groups and the lack of reliable or high-resolution data for consanguinity, we chose not to include it in our main calculations.

## Summarising the current evidence-base for α-thalassaemia gene frequency in Southeast Asia

Cartographic representations of the identified prevalence surveys were generated using ArcGIS 10.4.1 (ESRI Inc, Redlands, CA, USA). The descriptive maps reflect the spatial distribution of the prevalence surveys, along with their respective sample sizes and observed $\alpha^0$-, $\alpha^+$- and $\alpha^{ND}$-thalassaemia allele frequencies. Other features of the database, including the temporal distribution of the surveys, the identity of the populations studied (e.g. community, pregnant women, newborns, etc.) and the contribution of local Thai surveys to the evidence-base, were also examined.

## Mapping α-thalassaemia genetic diversity

Maps of the genetic diversity of α-thalassaemia across Southeast Asia were also generated (*Figure 6*). Given the heterogeneity in the reporting of different α-thalassaemia genotypes, we divided the genetic diversity data into two subtypes: (i) those surveys that only distinguished between the different α-thalassaemia forms ($\alpha^0$-, $\alpha^+$-, and $\alpha^{ND}$-thalassaemia), and (ii) those surveys that contained detailed count data for a range of common mutations. We focused on the 11 mutations that are most commonly reported in Southeast Asia or are part of standard multiplex polymerase chain reaction (PCR) methods: $-\alpha^{3.7}$, $-\alpha^{4.2}$, $\_^{SEA}$, $\_^{THAI}$, $\_^{MED}$, $\_^{FIL}$, $-(\alpha)^{20.5}$, Hb Adana (HBA2:c.179G > A), Hb CS (HBA2:c.427T > C). Hb Paksé (HBA2:c.429A > T), Hb Quong Sze (HbA2:c.377T > C) (*Liu et al., 2000*). An 'Other' category was used for other, rarer α-thalassaemia mutations. For the first data subtype, only surveys that tested for all three α-thalassaemia forms were mapped and the relative proportions of the different forms in the study sample were displayed using pie charts. For the latter, the same approach to that used in *Howes et al. (2013)* was used; the variant proportions were displayed using bar charts in which all variants that were explicitly tested for were included on the *x*-axis (*Howes et al., 2013*). This allowed information regarding the suite of variants that were tested for in the survey to be displayed as well as unambiguous representation of the absence of a variant in the study sample.

## Additional information

### Funding

| Funder | Grant reference number | Author |
| --- | --- | --- |
| European Research Council | 268904 | Carinna Hockham<br>Sunetra Gupta |

The funders had no role in study design, data collection and interpretation, or the decision to submit the work for publication.

### Author contributions

Carinna Hockham, Conceptualization, Data curation, Formal analysis, Visualization, Methodology, Writing—original draft; Supachai Ekwattanakit, Conceptualization, Data curation, Writing—review and editing; Samir Bhatt, Formal analysis, Methodology, Writing—review and editing; Bridget S Penman, Formal analysis, Supervision, Writing—review and editing; Sunetra Gupta, Supervision, Writing—review and editing; Vip Viprakasit, Conceptualization, Writing—review and editing; Frédéric B Piel, Conceptualization, Formal analysis, Supervision, Methodology, Writing—review and editing

Author ORCIDs

Carinna Hockham https://orcid.org/0000-0003-2126-5350
Sunetra Gupta http://orcid.org/0000-0002-9775-4006
Frédéric B Piel https://orcid.org/0000-0001-8131-7728

Decision letter and Author response

Decision letter https://doi.org/10.7554/eLife.40580.033
Author response https://doi.org/10.7554/eLife.40580.034

## Additional files

### Supplementary files

• Supplementary file 1. References for allele frequency data.
DOI: https://doi.org/10.7554/eLife.40580.022

• Supplementary file 2. References for genetic variant data.
DOI: https://doi.org/10.7554/eLife.40580.023

• Transparent reporting form
DOI: https://doi.org/10.7554/eLife.40580.024

### Data availability

All data generated or analysed during this study are included in the manuscript and supporting files. Source data files have been provided for Figures 1 and 3-6 and Figure 1—figure supplement 2.

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

# Appendix 1

DOI: https://doi.org/10.7554/eLife.40580.025

## Constructing a geodatabase of α-thalassaemia prevalence and/or genetic diversity surveys

### Library assembly

Three online biomedical literature databases (PubMed, ISI Web of Knowledge and Scopus) were systematically searched for all records referring to 'α-thalassaemia', 'alpha-thalassaemia', 'α-thal' or alpha-thal'. Broad search terms were used to make the search as inclusive as possible. Retrieved articles were imported into the bibliographic database, Endnote X7 (Thomson Reuters, Carlsbad, CA, USA) and grouped by country. The countries included in this study are shown in *Figure 1—figure supplement 1*. The last search was conducted on 21 July 2017. In addition, a non-systematic review of surveys available in local Thai journals was performed.

### Survey inclusion criteria

The title and abstract of each reference was reviewed for its suitability to this study and those that were considered not to be so were excluded from further review. This included animal studies, review articles and studies performed elsewhere in the world. The remaining references had their full texts (where available) reviewed using a set of strict inclusion criteria, which varied slightly depending on the type of data being reported (i.e. prevalence data or genetic diversity data).

For prevalence data, details on the number of individuals sampled and the different α-thalassaemia alleles and/or genotypes identified were needed. Thus, surveys that reported allele or genotype frequency without the sample size were excluded as were those that reported an overall α-thalassaemia gene frequency without distinguishing between different α-thalassaemia forms or genotypes. Given the important clinical differences between $\alpha^+$-, $\alpha^0$- and $\alpha^{ND}$-thalassaemia, many of the surveys focused on specific genotypes, in particular those containing an $\alpha^0$-thalassaemia allele (i.e. $-/\alpha\alpha$, $-/-$, $-\alpha/-$ and $\alpha\alpha^{ND}/-$). As a result, allele frequencies of the other α-thalassaemia forms were not always a reliable representation of their true allele frequency in the study population, despite them being tested for. In this instance, the reported frequencies of the alleles of secondary interest were entered into the database as missing values. Similarly, if an α-thalassaemia allele was not tested for, the survey was still included but the frequency value of the untested allele was entered as missing. For surveys where the diagnostic algorithm and genotype reporting were sufficiently detailed (e.g. all seven major variants tested using PCR and a range of genotypes reported), we assumed that the lack of explicit reporting of certain genotypes reflected a zero frequency of them. Data on the absence of α-thalassaemia in the studied populations were also included. Where necessary, allele frequencies were derived from their respective genotype frequencies by assuming Hardy-Weinberg proportions.(*Hardy, 1908*; *Weinberg, 1908*)

To be truly representative of the general population, surveys should take place in the community using random sampling. However, it became apparent early in the review process that very few surveys were conducted in this way. We therefore chose to include surveys that sampled from any of the following population groups, provided sampling was random, consecutive or universal: (i) the general community, (ii) pregnant women attending antenatal care and/or their husbands, (iii) cord blood and/or neonates in the absence of any inclusion/exclusion criteria or initial thalassaemia screening, (iv) individuals attending their yearly health check-up, and (v) opt-out volunteers, e.g. students, blood donors or army personnel. These groups were deemed to contain no inherent bias in α-thalassaemia prevalence and thus suitable for the purpose of this study. By contrast, surveys involving opt-in sampling, relatives of known cases of α-thalassaemia, putative anaemic cases, patients or individuals possessing a

β-globin variant (i.e. β-thalassaemia, β$^E$ or β$^S$) were excluded. Moreover, surveys that were carried out in specific ethnic groups were only included if the population being investigated was representative of the study area. Representativeness was determined based on information provided in the reference.

Whilst a range of diagnostic methods are available for the diagnosis of α-thalassaemia, (*Fucharoen and Winichagoon, 2011*) we only included surveys in which molecular diagnostic methods were used (e.g. polymerase chain reaction (PCR)-based techniques and DNA sequencing).(*Old and Henderson, 2010*) This is because other methods are unreliable for characterising the precise α-thalassaemia genotype of an individual. For instance, haematological indices cannot distinguish between α$^+$-thalassaemia and the wild-type state nor between α$^0$-thalassaemia and iron deficiency anaemia (both of which lead to microcytosis). (*Fucharoen and Winichagoon, 2011*) Haemoglobin analysis by capillary electrophoresis (CE), high performance liquid chromatography (HPLC), isoelectric focusing (IEF) and citrate agar electrophoresis (CAE) are all techniques that have been used to identify the quantity of Hb Bart's, Hb H, Hb CS and other types of haemoglobin relevant to α-thalassaemia, and are comparable in performance.(*Alauddin et al., 2017*; *Mantikou et al., 2009*) However, accurate diagnosis of specific α-thalassaemia genotypes is not possible with these techniques and their performance is reduced when there is concomitant inheritance of β-thalassaemia.(*Chui, 2005*; *Alauddin et al., 2017*) Given that the objective of this study was to describe and map allele frequencies, the inclusion of non-molecular studies could affect the reliability of allele frequency estimates.

The majority of surveys that provided information on the genetic diversity of α-thalassaemia mutations were cross-sectional surveys in which there was no prior knowledge or screening of the α-thalassaemia status of the study sample. All of the same inclusion criteria as those described above were applied – that is: (i) a clearly reported sample size and allele or genotype frequencies, (ii) a representative sample, and (iii) molecular diagnosis. In other surveys, the underlying mutations among individuals known to have α-thalassaemia were characterised. To avoid potential bias towards more severe variants in these surveys, only those that took place outside of the hospital setting and in the absence of any inclusion or exclusion criteria pertaining to disease severity were included. Surveys amongst other patient groups, for example β-thalassaemics, were also excluded. Again, only those surveys using molecular diagnostic methods were included.

## Georeferencing

To capture fine-scale spatial heterogeneities in α-thalassaemia gene frequency and/or genetic diversity, included surveys were mapped to the highest spatial resolution possible based on the information provided in the article. Online geopositioning gazeteers were used to identify the latitude and longitude decimal degree coordinates of the study location. For studies that could be georeferenced to the province or district level only, the centroid of the polygon was extracted from ArcMap 10.4.1 (ESRI Inc, Redlands, CA, USA) and used. For studies that were conducted across multiple locations but reported as a single frequency estimate, the coordinates for each site were obtained from the online gazeteers and the centroid of all the sites calculated using ArcMap. In some cases, this resulted in a data point that fell between two land areas, for example Peninsular Malaysia and Malaysian Borneo.

## Appendix 2

DOI: https://doi.org/10.7554/eLife.40580.025

# Generation of continuous allele frequency maps for Thailand

Continuous maps of the allele frequencies of $\alpha^0$-, $\alpha^+$- and $\alpha^{ND}$-thalassaemia were generated for Thailand only. This is because there was considerably more data for Thailand than for any of the other countries in the region, in part due to the inclusion of data from local journals. Data from Thailand and all of its neighbouring countries (Myanmar, Lao PDR, Cambodia and Malaysia) were used in this part of the analysis to preclude the possibility of a border effect on the predicted allele frequencies. Surveys that were reported only at the national level were excluded. To generate three separate maps of $\alpha$-thalassaemia allele frequency, data on each $\alpha$-thalassaemia form were extracted from the database and used as input for three separate models. The observed allele frequency data were transformed through an empirical logit, which, for databases $\geq$ 20 surveys, can be well approximated by a Gaussian likelihood. (**Diggle and Ribeiro, 2007**)

The goal of the geostatistical analysis was to estimate model parameters and generate predictions of $\alpha$-thalassaemia allele frequencies in Thailand at fine spatial resolution. We employed, using the 'r-inla' package, an easily available Bayesian geostatistical framework involving a stochastic partial differential equation (SPDE) approach(**Lindgren et al., 2011**) with an integrated nested Laplace approximation (INLA)(**Rue et al., 2009**) algorithm for Bayesian inference. The theoretical principles of this approach have been described in detail elsewhere. (**Lindgren et al., 2011**; **Rue et al., 2017**; **Cameletti et al., 2013**) A brief overview is provided below.

Let $Y$ be the observed data, which in this study is the set of allele frequency estimates in Thailand and neighbouring countries. Each value $y_i$ represents the allele frequency $y$ at location $i$. In general, we can assume that $Y$ is generated by an underlying Gaussian process, denoted as $S(x)$, such that any evaluations of $S(x)$ are multivariate normal distributed with a given mean and covariance function. Therefore, $S(x)$ can be interpreted as a continuously indexed Gaussian field (GF), whereby the effect at each location has a multivariate normal as

$$p(y|x,\theta) \sim \prod_{i=1}^{n} MVN(y_i|x_i,\theta)$$

where, $y$ denotes allele frequency, $x$ is the Gaussian random effect on $y$, $n$ represents the number of spatially unique data points, $i$ represents location and $\theta$ denotes the hyperparameters of $x$ (i.e. mean $\mu$ and dispersion parameter $k$). Observations $y$ are assumed to be conditionally independent, given $x$ and $\theta$.

The first law of geography states that: 'everything is related to everything else, but near things are more related to distant things."(**Tobler, 1970**) This phenomenon is termed spatial dependency, or autocorrelation, and its inclusion in spatial models can greatly improve predictive performance. Several methods for defining the covariance between spatial points exist; here we use the Matérn class of covariance function in which the correlation function is defined based on the Euclidean distance between locations.(**Krainski et al., 2017**) Specifically, the stationary Matérn covariance function assumes that if we have two pairs of points separated by the same Euclidean distance $h$, both pairs have same correlation, with correlation monotonically decreasing as a function of $h$.

Matrix algebra operations on a continuously indexed GF are computationally intensive to model. To overcome this, we use a finite element solution of an SPDE approximation to the Matérn covariance function, resulting in a finite dimensional Gaussian Markov random field (GMRF). The finite element representation represents $S(x)$ by means of a basis function representation defined on a triangulation of the domain under study, represented as: (**Lindgren et al., 2011**)

$$S(x) = \sum_{g=1}^{G} \psi_g(x) S_g,$$

where, *G* is the number of vertices in the triangulation, $\psi_g(\cdot)$ *are piecewise polynomial basis functions on each triangle* (**Moraga et al., 2015**). The SPDE approach replaces the Matérn covariance function's dense covariance matrix by a sparse reduced rank matrix. Following the generation of the GMRF prior, the posterior distribution is approximated by an Integrate Nested Laplace Approximation (INLA), which uses a combination of analytical approximation and numerical integration methods to approximate the posterior distribution at each point in the GMRF. The model's fully Bayesian hierarchical formulation is as follows:

$$\langle Y_i | S, \theta \rangle \sim p \langle Y_i | S, \theta \rangle,$$

$$\langle S | \theta \rangle \sim N\left(0, Q(\theta)^{-1}\right),$$

$$(\theta) \sim p(\theta),$$

where, again, Y denotes allele frequency (the observation variable), S denotes the underlying Gaussian field, θ denotes a vector of hyperparameters and Q is the sparse precision matrix.

One hundred conditional simulations of the model were performed, whereby all of the pixels in the map were jointly simulated such that spatial autocorrelation between the pixels was accounted for.(**Gething et al., 2010**) This generated PPDs of allele frequency for each pixel in a 1 km x 1 km grid of Thailand, which were then used to calculate the mean and 95% Bayesian credible intervals for the predicted allele frequencies.

Given the relatively small number of surveys in each dataset, the predictive ability of the model for each form of α-thalassaemia was assessed using a 10-fold cross validation procedure. For each form of α-thalassaemia, the original dataset was randomly partitioned into 10 equal-sized data subsets. In each of 10 separate cross-validation experiments, one of the data subsets was retained as the test set whilst the remaining nine data subsets were used as the training set. The disparity between the model predictions and the observed allele frequencies in the test set was quantified for each cross-validation iteration using: (i) the mean absolute error (MAE), defined as the average magnitude of errors in the predicted values, and (ii) the correlation. The average across the ten cross-validation experiments was then calculated.

R code for the geostatistical model was generated by SB and is available on request. The model was implemented in R using the 'r-inla' package, while the predicted allele frequency and disease burden maps were generated in ArcMap.

## Appendix 3

DOI: https://doi.org/10.7554/eLife.40580.025

### Generating newborn estimates of Hb Bart's hydrops fetalis and HbH disease for Thailand

Newborn estimates of Hb Bart's hydrops fetalis and HbH disease (deletional and non-deletional) were generated by pairing our predicted allele frequency PPDs with population and birth rate data. First, the number of births in each pixel of a 1 km x 1 km grid of Thailand were calculated by multiplying high-resolution 2020 population count data obtained from the WorldPop project website (www.worldpop.org.uk) with the national birth rate. In the UN World Population Prospects 2017 Revision, birth rates were provided as low-, medium- and high-fertility variant projections for 5 year periods. To obtain 2020 estimates of the number of births in Thailand, we used the average of the two 5 year periods 2015–2020 and 2020–2025.

One-hundred realisations of the geostatistical models for $\alpha^0$-, $\alpha^+$- and $\alpha^{ND}$-thalassaemia allele frequency were run in parallel. For each realisation of the models, the estimated genotype frequencies for Hb Bart's hydrops fetalis disease (–/–), deletional HbH disease (-$\alpha$/–) and non-deletional HbH disease ($\alpha\alpha^{ND}$/–) were calculated using the predicted allele frequencies, assuming Hardy-Weinberg proportions for a four-allele system (see *Equation 1* in the main text). We also examined the effect that the inclusion of consanguinity into the Hardy-Weinberg equations would have on the estimates (see *Table 1* in the main text). Genotype frequencies were then multiplied by the number of births in the corresponding 1 km x 1 km pixel of the birth count map described above. A PPD for the number of births in each pixel was therefore generated, from which point estimates and uncertainty around these estimates were computed.

The lower bounds of our credible intervals were calculated using the 2.5th percentile for genotype frequency estimates and the low-fertility variant for birth count, whilst the higher bounds were calculated using the 97.5th percentile for genotype frequency estimates and birth count data based on the high-fertility variant.

In order to compare the geostatistical method employed in this study with methods used to generate previous $\alpha$-thalassaemia disease estimates, we obtained 2003 population count data from the Global Rural-Urban Mapping Project (GRUMP) and paired this with the same birth rate as that used in Modell and Darlison (17 births per 1000 population; *Modell and Darlison, 2008*). The same method described above was used; however, no uncertainty intervals were calculated as no corresponding low- and high-fertility variants for birth rate were reported in their study.

## Appendix 4

DOI: https://doi.org/10.7554/eLife.40580.025

# Generation of maps of genetic variation of α-thalassaemia

From a clinical perspective, the most important distinction between different α-thalassaemia alleles is the number of α-globin genes affected and whether the mutation is of the deletional or non-deletional type – i.e. the distinction between $\alpha^0$-, $\alpha^+$- and $\alpha^{ND}$-thalassaemia.(**Piel and Weatherall, 2014**) However, molecular diagnosis of α-thalassaemia is mutation-specific and as such the profile of α-thalassaemia mutations that are found in a population determines which mutations are tested for in screening programmes. A better understanding of the genetic diversity patterns of α-thalassaemia at the mutation level therefore has important implications for programme design.(**Old and Henderson, 2010**; **Galanello, 2013**) In this study, we include surveys that report on genetic diversity at both levels of specificity.

For the descriptive analysis of the relative proportions of each of the three major forms of α-thalassaemia ($\alpha^0$-, $\alpha^+$ and $\alpha^{ND}$-thalassaemia), only surveys which tested for all three forms were included. This was to avoid misinterpretation of untested forms as being absent. The relative proportions of the different forms were represented using pie charts for which the denominator was the number of α-thalassaemia alleles in the study. The size of the pie charts reflects sample size on the log scale to allow clear visualisation of the data points. A jitter of 0.5-1° latitude and longitude decimal degree coordinates was applied to each survey to avoid the overlapping of surveys carried out in the same location.

The subset of surveys that reported on specific α-thalassaemia mutations indicated that 11 variants were commonly tested for in the Southeast Asia region: single deletion mutations, -$\alpha^{3.7}$ and -$\alpha^{4.2}$; non-deletion mutations, Hb Adana, Hb Constant Spring, Hb Paksé and Hb Quong Sze; and double deletion mutations, (α)20.5, FIL, MED, SEA and THAI. Other non-deletion mutations such as Hb Q-Thailand and initiation codon mutations were tested for in very few surveys ($n = 13$) and were grouped together into a single category as 'Other'. Surveys were mapped spatially using bar charts to depict variant proportions. All variants that were tested for in a survey were represented along the x-axis of the corresponding chart. Empty spaces are therefore indicative of the variant being absent in the sample. To allow clear visualisation of the bar charts, data for Thailand were displayed separately, whilst the remaining Southeast Asian countries were divided into two maps: (i) Myanmar, Lao PDR, Cambodia and Vietnam in the north, and (ii) Malaysia, Indonesia and Singapore in the south. Bar charts could not be placed at their precise geographical location and so surveys were mapped as points and connected to their corresponding bar chart by a black line.

# Appendix 5

DOI: https://doi.org/10.7554/eLife.40580.025

## Existing α-thalassaemia maps and characteristics of α-thalassaemia database

Among the surveys included in the final geodatabase, $\alpha^0$-thalassaemia was the most extensively studied form ($n$ = 97), followed by $\alpha^{ND}$-thalassaemia ($n$ = 50) and then $\alpha^+$-thalassaemia ($n$ = 47). The majority of studies (95.3%) were carried out after 1995 and in particular from 2005 onwards (78.3%) (*Figure 1—figure supplement 2*). In terms of the population groups sampled, 21.7% of surveys were community-based, 59.4% were carried out amongst selected but unbiased population groups (e.g. pregnant women, husbands of pregnant women, neonates and cord blood) and 8.5% in opt-out volunteers. One (<1%) survey providing only genetic diversity data was carried out in individuals known to have α-thalassaemia. Twelve per cent of surveys provided no detailed information on the sampling methodology but were judged to be unbiased (based on other information provided in the article or personal communication with the corresponding author) and were included. Sample size ranged from 4 to 55,796. Small sample sizes (e.g. $n$ = 4) came from surveys that were carried out across multiple geographic locations and the α-thalassaemia frequencies reported separately for each. *Appendix 5—table 1* provides a breakdown of the survey features for the overall database and for prevalence surveys and genetic variant surveys, separately.

**Appendix 5—table 1.** Summary of the α-thalassaemia dataset characteristics according to the type of data provided (allele frequency data or genetic variant data), and overall. Numbers correspond to individual surveys that met the study inclusion criteria. As some sources reported more than one survey from multiple locations or in multiple population groups, the number of surveys is greater than the number of references in the *Supplementary files 1* and *2*. Some surveys reported data on both α-thalassaemia prevalence and genetic diversity and are therefore included twice in these columns, but once in the overall column.

| | Allele frequency data | Genetic diversity data | Overall |
|---|---|---|---|
| Total surveys | 104 | 60 | 106 |
| Number of countries | 8 | 8 | 8 |
| Publication time | | | |
| 1959–1969 | 0 | 0 | 0 |
| 1970–1979 | 0 | 0 | 0 |
| 1980–1989 | 2 | 2 | 2 |
| 1990–1999 | 8 | 5 | 8 |
| 2000–2009 | 46 | 15 | 47 |
| 2010–2017 | 47 | 38 | 48 |
| N/A | 1 | 0 | 1 |
| Spatial extent | | | |
| Admin 0 centroids | 4 | 4 | 4 |
| Admin one centroids | 27 | 12 | 27 |
| Admin two centroids | 10 | 8 | 9 |
| Admin three centroids | 3 | 3 | 3 |
| Points | 46 | 22 | 47 |
| Multiple centroids/points | 14 | 11 | 16 |
| Total individuals sampled | 132,157 | 32,237 | 133,649 |

*Appendix 5—table 1 continued on next page*

*Appendix 5—table 1 continued*

|  | Allele frequency data | Genetic diversity data | Overall |
|---|---|---|---|
| Survey count by sample size |  |  |  |
| ≤50 | 3 | 2 | 4 |
| 51–250 | 22 | 18 | 22 |
| 251–500 | 38 | 23 | 38 |
| 501–750 | 21 | 10 | 21 |
| 751–1000 | 3 | 1 | 3 |
| >1000 | 17 | 6 | 18 |

DOI: https://doi.org/10.7554/eLife.40580.030

**Appendix 5—table 2.** Observed allele frequency ranges for different α-thalassaemia forms.

| Country | Allele frequency range (%) | | |
|---|---|---|---|
|  | $\alpha^0$-thalassaemia | $\alpha^+$-thalassaemia | $\alpha^{ND}$-thalassaemia |
| Brunei Darussalam | No surveys identified | No surveys identified | No surveys identified |
| Cambodia | 0.80–1.10 | 10.30–26.30 | 2.44–4.20 |
| Indonesia | No surveys identified | 2.91 | No surveys identified |
| Lao PDR | 0.00–6.19 | 4.60–40.00 | 2.28–9.00 |
| Malaysia | 0.00–1.92 | 0.00–16.80 | 0.00–16.25 |
| Myanmar | 0.93 | 20.58 | No surveys identified |
| Philippines | No surveys identified | No surveys identified | No surveys identified |
| Singapore | 0.86–0.90 | 1.88–3.04 | 0.04 |
| Thailand | 0.00–9.29 | 2.98–21.43 | 0.00–7.30 |
| Vietnam | 0.00–2.66 | 1.59–14.4 | 2.07–14.43 |

DOI: https://doi.org/10.7554/eLife.40580.031

