## [Decision Letter]

[**Editorial note:** This article has been through an editorial process in which the authors decide how to respond to the issues raised during peer review. The Reviewing Editor's assessment is that all the issues have been addressed.]

Thank you for submitting your article "Estimating the burden of α-thalassaemia in Thailand using a comprehensive prevalence database for Southeast Asia" for consideration by *eLife*. Your article has been reviewed by two peer reviewers, and the evaluation has been overseen by a Reviewing Editor and Eduardo Franco as the Senior Editor. The two reviewers have opted to remain anonymous.

The Reviewing Editor has highlighted the concerns that require revision and/or responses, and we have included the separate reviews below for your consideration. If you have any questions, please do not hesitate to contact us.

Summary:

This is important, novel work. The analyses show a visualized map of α-thalassemia prevalence and genotypic diversity, which can be of a large benefit for future prevention policy in Thailand. A major strength of this work is an extensive review of relevant prevalence and genotype reports. Overall, this represents a valuable and much-needed contribution to the literature about this neglected genetic disorder.

Major concerns:

The model's smoothing effect appears to be quite strong, avoiding the more extreme points such as the 7% frequency of ND-thalassaemia quoted in paragraph four of subsection “Overall distribution of α-thalassaemia across Southeast Asia”. This ND-thalassaemia model also had relatively low correlation statistics. The authors need to discuss this further, and why the range in the predicted mean map is so homogenous for the ND-thalassaemias?

Figures 4-6: why did you select to map proportions and not frequencies? Frequencies would seem a more informative metric. These maps show considerable variation between surveys very close together geographically (e.g. Vietnam and peninsular Malaysia), which would merit some mention in the Discussion.

The authors need to clarify if this model underestimates the burden of α-thalassemia diseases in Thailand due to the overlapping prevalence of α- and β-thalassemia.

The authors should clarify why Bangkok is predicted to have the highest burden of both hydrops fetalis and HbH disease (deletional and non-deletional), despite not having the highest allele frequency of α0-thalassemia (Figure 1).

The influence of consanguinity is interesting. Authors ran their predictions for 2003 both with and without an adjustment coefficient, finding a relatively small difference. Can the authors explain their decision to not include this adjustment factor in their main predictions for 2020?

Separate reviews (please respond to each point):

*Reviewer #1:*

The highlight of this work was, for the first time, employment of geostatistical model to estimate burden of severe form of α-thalassemia in Thailand, and to the lesser extent in other Southeast Asian countries, in 2020. The analyses also show visualized map of α-thalassemia prevalence and genotypic diversity, which can be of a large benefit for future prevention policy in Thailand.

The strength of this work is an extensive review of relevant prevalence and genotype reports, not only from those published in international database, but also from local journals. Criteria for selection of unbiased surveys was clearly stated in the supplementary method 1. The limitation of this work was that these maps represent a prediction or estimation of prevalence. Therefore accuracy of the estimation depends largely upon availability of published data from individual area of the region. The authors, however, clearly stated this limitation in the discussion. Overlapping prevalence of α- and β-thalassemia are also commonly observed in Thailand. This includes e.g., AEBart's disease or AEBart's with CS, clinical phenotype of which can be more severe compare to those with deletional or non-deletional HbH alone. Therefore this model remain underestimates burden of α-thalassemia diseases in Thailand in this context. Future work on model of β-thalassemia prevalence will be very helpful.

Other specific comment:

In the Result subsection “Estimates of number of affected newborns in Thailand”: The authors should clarify why Bangkok is predicted to have highest burden of both hydrops fetalis and HbH disease (deletional and non-deletional), despite not having the highest allele frequency of α0-thalassemia (Figure 1).

*Reviewer #2:*

Hockham and colleagues present an impressive description of the available data about the burden of α-thalassaemia in Thailand and the broader ASEAN region. They make a strong case for the need for this analysis, and then detail a very thorough description of the available information, together with modelled surfaces of α-thalassaemia allele frequencies in Thailand. The scarcity of evidence outside Thailand constrained the modelling analysis to this country alone. From these maps, estimates of newborns affected by different forms of α-thalassaemia in Thailand are projected for 2020. Overall, this presentation of the evidence, and extension thereof through the modelling analysis, represents a valuable and much-needed contribution to the literature about this neglected genetic disorder – congratulations to the authors on this commendable undertaking and their clear distillation of a highly complex disorder.

The Introduction also quotes frequencies of close to 80% in Southeast Asia (citing reference 1), while values of this magnitude are not discussed in the section "Overall distribution of α-thalassaemia across Southeast Asia". If the survey reporting 80% did not meet the inclusion criteria for the present study, perhaps take out from the Introduction as could be misleading and confusing.

Supplementary methods 1, Survey inclusion criteria: the maps risk being misleading if ethnic groups indigenous to an area but not representative are included, given they may today be a minority group. Were any such surveys in fact included in the mapping dataset?

Figures 1 and 2 need to be swapped around (data points first, then the continuous maps).

Does the Ministry of Health reporting system include any statistics on α-thalassaemia disorders that could be compared to? Or are the necessary diagnostics not sufficiently accessible for figures to be helpful?

Minor Comments:

Introduction paragraph three: brief reference to the variation in the ND phenotypes would be helpful. E.g. "non-deletional variants are variable and typically associated with…"

Reference 22: Author name is "Darlison", not "Darlinson"

Reference 25 (Introduction paragraph five): this reference contains no geostatistical modelling. Suggest citing this instead: https://www.ncbi.nlm.nih.gov/pubmed/23152723

Results section paragraph two: was a minimum sample size imposed? (maybe c/ref to the supplementary files here)

Subsection “Estimates of number of affected newborns in Thailand”: what limits were used to define the spatial extent of Bangkok?

Subsection “Overall distribution of α-thalassaemia across Southeast Asia”: Supplementary file 2 doesn't currently include this data.

Subsection “Genetic diversity of α-thalassaemia across Southeast Asia”: "only distinguished between them" is not very clear; could change to something like "and maps relative proportions of each allele type without giving specific variant details".

Discussion, line three: suggest some additional detail in relation to the "epidemiological transition" – such as adding "from infectious to non-communicable causes of disease and death".

Discussion, paragraph one, last sentence: "high prevalence" is difficult to see in the maps in Figure 2. Perhaps adjust the wording.

Discussion, paragraph two, final sentence: are there no national statistics on the numbers of stillbirths?

Discussion, subsection “Model strengths and limitations”, third paragraph: malaria is listed as a possible influence of allele frequencies – should this be "historic" rates of malaria? Given that the diagnostics used here are molecular, this should not have any haematological impact of malaria infection on the diagnostic outcomes.

Subsection “Refining estimates of the annual number of neonates affected by severe disease forms”: state the value of the consanguinity coefficient, and list its source. A reference to explain how F is applied in the Hardy-Weinberg equations would be useful – e.g.: https://www.ncbi.nlm.nih.gov/pubmed/23950147

Figure 2: the specific locations of the points in panel A are hard to see, but would be helpful in exploring the modelled surfaces. Perhaps include a simpler map just showing the survey locations in the Supplementary information for a comparison with the modelled outputs. Or try scaling the data to resize the points?

Figure 2: would also include the star meaning in the text legend.

Figure 3: Legend needs to include some reference sample size pie charts; the pie charts are also quite difficult to distinguish by size, possible to widen the size range?

Figure 7: the model input indicates that "transformed allele frequency" was input into the model – what transformation was applied and why?

---

## [Author Response]

Major concerns:The model's smoothing effect appears to be quite strong, avoiding the more extreme points such as the 7% frequency of ND-thalassaemia quoted in paragraph four of subsection “Overall distribution of α-thalassaemia across Southeast Asia”.

We have carefully rechecked the model’s smoothing effect in light of this comment. The smoothing effect reflects both the quantity, size and heterogeneity of surveys included in our study. To expand on this, we have provided more details in paragraph three in the Model strengths and limitations section in the Discussion as follows:

“Spatial smoothing is an important component of our geostatistical model. For the modelling approach used here, the range function (i.e. the extent of spatial autocorrelation) is defined by a parameter within the SPDE framework and takes a prior distribution. The smoothing in the approximate posterior therefore balances over- and under-fitting and is necessary to ensure that the model predicts adequately without fitting the idiosyncrasies of the data. As a result, the model does not predict allele frequencies that fully reflect heterogeneity between nearby surveys. Although extensive variation in allele frequencies between different ethnic groups in similar geographic locations has been observed in Thailand (36) and other countries (e.g. Sri Lanka), this could not be reflected in our predicted allele frequencies. For example, allele frequencies of around 3.65% for the Hb CS mutation have been reported in the Khmer ethnic group in Surin and Buriram provinces, whilst our model predicts maximum allele frequencies of 1.65% here. This smoothing process can similarly explain why the highest observed αND-thalassaemia frequency of 7% in Khon Kaen was not reproduced in the predicted maps; the smoothing process is also taking into account the presence of multiple nearby surveys of larger sample size reporting lower frequencies, thereby masking the extreme value of 7%.”

This ND-thalassaemia model also had relatively low correlation statistics. The authors need to discuss this further, and why the range in the predicted mean map is so homogenous for the ND-thalassaemias?

We spent a substantial amount of time exploring alternative models for the ND-thalassaemia map, for example Gaussian models that used a higher number of nodes in the triangular mesh as well as models that used a betabinomial distribution for the observed data. However, the original model presented in the manuscript remained the best-performing model. We have added a few sentences to the end of the above Discussion paragraph to comment on the homogeneity in the predictions and the low correlation statistic as follows:

“In fact, our model-based predictions for αND-thalassaemia are remarkably homogenous and the average correlation between the observed and predicted frequencies is low (0.47). This is because the close-range heterogeneity in the observed data, coupled with the absence of a long-range trend in frequency (as is observed for α0- and α+-thalassaemia), makes it difficult for the model to discern a signal.”

Figures 4-6: why did you select to map proportions and not frequencies? Frequencies would seem a more informative metric.

We have mapped both the proportions (Figure 3) and the frequencies (Figures 4–6), and corrected the legends accordingly. We have also amended the main text to reduce any ambiguity as follows:

“Figure 3 (Figure 3—source data 1) displays surveys that included all three α-thalassaemia forms (α0-, α+- and αND-thalassaemia), allowing relative proportions of each of the forms to be calculated without giving specific variant details.”

These maps show considerable variation between surveys very close together geographically (e.g. Vietnam and peninsular Malaysia), which would merit some mention in the Discussion.

We fully agree and we have added the following paragraph under the section Patterns of genetic variation and their public health implications in the Discussion:

“The pattern of genetic diversity observed in this study indicates variable distributions of mild and severe α-thalassaemia forms. Reasons for this are unclear. However, high variant heterogeneity has been observed for other genetic disorders (e.g. G6PD deficiency) in Southeast Asia, (34) which might suggest a similar underlying cause. In their global study, Howes et al. noted that G6PD variants were most diverse in East Asia and the West Pacific, where *P. falciparum* parasites show strong population structure with lower genetic relatedness between populations in the region. Indeed, *P. falciparum* has been shown to display genetically structured populations within Thailand alone. (35) It is possible that the evolutionary dynamics between *P. falciparum* and haemoglobin variants, including α-thalassaemia, are more complex than we currently appreciate.”

The authors need to clarify if this model underestimates the burden of α-thalassemia diseases in Thailand due to the overlapping prevalence of α- and β-thalassemia.

Although this was briefly mentioned in the Discussion under Model strengths and limitations, we have amended the sentence to clarify, as suggested:

“Furthermore, the estimates presented here do not include compound disorders, such as EA Bart’s and EF Bart’s diseases (HbH disease with heterozygous and homozygous forms of βE, another clinically important structural β-globin variant, respectively), and therefore remain underestimates of the overall burden of α -thalassaemia disorders in Thailand.”

The authors should clarify why Bangkok is predicted to have the highest burden of both hydrops fetalis and HbH disease (deletional and non-deletional), despite not having the highest allele frequency of α0-thalassemia (Figure 1).

To clarify this point, we have amended the relevant sentence in the Results section as follows:

“The highest absolute burden of hydrops fetalis is predicted in Bangkok city (57 [CI: 13 – 151]), with its high population density, and Udon Thani (23 [CI: 6 – 66]) in the northeastern zone, where some of the highest α0-thalssaemia allele frequencies are predicted.”

The influence of consanguinity is interesting. Authors ran their predictions for 2003 both with and without an adjustment coefficient, finding a relatively small difference. Can the authors explain their decision to not include this adjustment factor in their main predictions for 2020?

This is a recurring challenge associated with consanguinity data. We have added the following clarification to the Materials and methods section:

“We used the online global database of consanguinity estimates (www.consang.net) to identify an upper limit for the coefficient of consanguinity for Thailand (F = 0.01) (52). However, due to important variations of this coefficient between ethnic groups and the lack of reliable or high-resolution data for consanguinity, we chose not to include it in our main calculations.”

Separate reviews (please respond to each point):

Reviewer #1:

[…] The strength of this work is an extensive review of relevant prevalence and genotype reports, not only from those published in international database, but also from local journals. Criteria for selection of unbiased surveys was clearly stated in the supplementary method 1. The limitation of this work was that these maps represent a prediction or estimation of prevalence. Therefore accuracy of the estimation depends largely upon availability of published data from individual area of the region. The authors, however, clearly stated this limitation in the discussion. Overlapping prevalence of α- and β-thalassemia are also commonly observed in Thailand. This includes e.g., AEBart's disease or AEBart's with CS, clinical phenotype of which can be more severe compare to those with deletional or non-deletional HbH alone. Therefore this model remain underestimates burden of α-thalassemia diseases in Thailand in this context. Future work on model of β-thalassemia prevalence will be very helpful.Other specific comment:In the Result subsection “Estimates of number of affected newborns in Thailand”: The authors should clarify why Bangkok is predicted to have highest burden of both hydrops fetalis and HbH disease (deletional and non-deletional), despite not having the highest allele frequency of α0-thalassemia (Figure 1).

As described above, we have amended the relevant sentence in the Results section as follows: “The highest absolute burden of hydrops fetalis is predicted in Bangkok city (57 [CI: 13 – 151]), with its high population density, and Udon Thani (23 [CI: 6 – 66]) in the northeastern zone, where some of the highest α0-thalssaemia allele frequencies are predicted.”

Reviewer #2:

Hockham and colleagues present an impressive description of the available data about the burden of α-thalassaemia in Thailand and the broader ASEAN region. They make a strong case for the need for this analysis, and then detail a very thorough description of the available information, together with modelled surfaces of α-thalassaemia allele frequencies in Thailand. The scarcity of evidence outside Thailand constrained the modelling analysis to this country alone. From these maps, estimates of newborns affected by different forms of α-thalassaemia in Thailand are projected for 2020. Overall, this presentation of the evidence, and extension thereof through the modelling analysis, represents a valuable and much-needed contribution to the literature about this neglected genetic disorder – congratulations to the authors on this commendable undertaking and their clear distillation of a highly complex disorder.The Introduction also quotes frequencies of close to 80% in Southeast Asia (citing reference 1), while values of this magnitude are not discussed in the section "Overall distribution of α-thalassaemia across Southeast Asia". If the survey reporting 80% did not meet the inclusion criteria for the present study, perhaps take out from the Introduction as could be misleading and confusing.

Thank you very much for highlighting this. This was an error – in fact reference 1 indicates that these very high frequencies have been reported in India and Papua New Guinea, not in Southeast Asia. We have corrected this in the Introduction as follows: “It is estimated that up to 5% of the world’s population carries at least one α-thalassaemia variant, with some populations (e.g. in India and Papua New Guinea) reporting gene frequencies of close to 80% (1)”.

Supplementary methods 1, Survey inclusion criteria: the maps risk being misleading if ethnic groups indigenous to an area but not representative are included, given they may today be a minority group. Were any such surveys in fact included in the mapping dataset?

We agree that this sentence is misleading and have changed it as follows:

“Moreover, surveys that were carried out in specific ethnic groups were only included if the population being investigated was representative of the study area. Representativeness was determined based on information provided in the reference.”

Figures 1 and 2 need to be swapped around (data points first, then the continuous maps).

Thank you for pointing this out. We have swapped Figures 1 and 2 as suggested.

Does the Ministry of Health reporting system include any statistics on α-thalassaemia disorders that could be compared to? Or are the necessary diagnostics not sufficiently accessible for figures to be helpful?

Our Thai collaborators confirmed that statistics on severe α-thalassaemia cases born each year are not available for Thailand. Commonly reported figures are based on an old study from 1988 (Fucharoen and Winichagoon, 1988). Prenatal screening data from the Ministry of Health are not publicly accessed nor usable in their current format as they do not distinguish between severe forms of α-thalassaemia and β-thalassaemia.

Minor Comments:Introduction paragraph three: brief reference to the variation in the ND phenotypes would be helpful. E.g. "non-deletional variants are variable and typically associated with…"

We have added the following sentence to the Introduction: “However, even amongst non-deletional variants, considerable phenotypic variability is observed (13).” We have also added a reference: Variable clinical phenotypes of α-thalassemia syndromes (Singer, 2009).

Reference 22: Author name is "Darlison", not "Darlinson"

Thank you for pointing this out. We have rectified the mistake.

Reference 25 (Introduction paragraph five): this reference contains no geostatistical modelling. Suggest citing this instead: https://www.ncbi.nlm.nih.gov/pubmed/23152723

Thank you for pointing this out. We have amended the reference as per your suggestion.

Results section paragraph two: was a minimum sample size imposed? (maybe c/ref to the supplementary files here)

We have referred the readers to the Supplementary Results section (now Appendix 5) for further details by adding the following statement in the main text “Further details on the final database are provided in Appendix 5.”

Subsection “Estimates of number of affected newborns in Thailand”: what limits were used to define the spatial extent of Bangkok?

When we refer to Bangkok, we are referring specifically to Bangkok city and not Bangkok metropolis, which includes some of the neighbouring provinces. I have changed all instances of “Bangkok” to “Bangkok city”. I have also included an extra supplement to Figure 2 (Figure 2—figure supplement 2) that provides a reference map of Thailand provinces and referred to the map in relevant places in the main text.

Subsection “Overall distribution of α-thalassaemia across Southeast Asia”: Supplementary file 2 doesn't currently include this data.

We have added all of the countries to the table. We have also amended the sentence to make it clearer that we are referring to the major α-thalassaemia forms and not specific variants: “Appendix 5—table 2 shows the range of allele frequencies observed for the different α-thalassaemia forms (α0-, α+- and αND-thalassaemia) in each country.”

Subsection “Genetic diversity of α-thalassaemia across Southeast Asia”: "only distinguished between them" is not very clear; could change to something like "and maps relative proportions of each allele type without giving specific variant details".

We have amended this section as follows to clarify:

“Figure 3 (Figure 3—source data 1) displays surveys that included all three α-thalassaemia forms (α0-, α+- and αND-thalassaemia), allowing relative proportions of each of the forms to be calculated without giving specific variant details. Figures 4–6 display surveys that provided information on the frequencies of specific α-thalassaemia variants (e.g. --SEA, -α3.7, etc.). For these, the variants that were tested for differ between surveys. Some surveys are included in both Figure 3 and Figures 4–6. For the latter figures, the region has been divided to improve visualisation of the data.”

Discussion, line three: suggest some additional detail in relation to the "epidemiological transition" – such as adding "from infectious to non-communicable causes of disease and death".

We agree that this may be helpful to some readers. We have amended the sentence as follows:

“α-thalassaemia is a neglected public health problem whose burden has, to date, been largely overlooked, but for which morbidity is expected to increase in the coming decades as a result of the epidemiological transition, whereby acute infectious disease is replaced by chronic disease as the predominant cause of morbidity and mortality.”

Discussion, paragraph one, last sentence: "high prevalence" is difficult to see in the maps in Figure 2. Perhaps adjust the wording.

We have adjusted the wording accordingly. The sentence now reads:

“This is particularly true for countries in Southeast Asia as well as the Mediterranean area, where severe forms of α-thalassaemia (i.e. α0-thalassaemia) are found.”

Discussion, paragraph two, final sentence: are there no national statistics on the numbers of stillbirths?

We have double-checked with our Thai collaborators and co-authors. The only reference that we could find is the one cited.

Discussion, subsection “Model strengths and limitations”, third paragraph: malaria is listed as a possible influence of allele frequencies – should this be "historic" rates of malaria? Given that the diagnostics used here are molecular, this should not have any haematological impact of malaria infection on the diagnostic outcomes.

We agree with this comment. We have changed the sentence to reflect this:

“It is likely that other factors influence the allele frequencies of the different α-thalassaemia forms, but have not been considered in this mapping study, including ethnicity, consanguinity, historic rates of malaria (both *Plasmodium falciparum* and *P. vivax*) (37) and population migration patterns.”

Subsection “Refining estimates of the annual number of neonates affected by severe disease forms”: state the value of the consanguinity coefficient, and list its source. A reference to explain how F is applied in the Hardy-Weinberg equations would be useful – e.g.: https://www.ncbi.nlm.nih.gov/pubmed/23950147

As suggested, we have added the value of the consanguinity coefficient used and a reference, in the Material and methods paragraph: The revised text is as follows:

“We used the online global database of consanguinity estimates (www.consang.net) to identify an upper limit for the coefficient of consanguinity for Thailand (F = 0.01) (53). However, due to the lack of reliable or high-resolution data for consanguinity, we chose not to include it in our main calculations.”

Figure 2: the specific locations of the points in panel A are hard to see, but would be helpful in exploring the modelled surfaces. Perhaps include a simpler map just showing the survey locations in the Supplementary information for a comparison with the modelled outputs. Or try scaling the data to resize the points?

We have included these maps in Figure 2—figure supplement 1 and referred to them in the appropriate section of the Results (subsection “Continuous allele frequency maps for Thailand” and in the legend for Figure 2).

Figure 2: would also include the star meaning in the text legend.

We have added the following sentence to the legend of Figure 2: “Surveys that could only be mapped at the national level are indicated by a black star.”

Figure 3: Legend needs to include some reference sample size pie charts; the pie charts are also quite difficult to distinguish by size, possible to widen the size range?

We have added reference sample pie charts to the legend and increased the range size. We have also removed the log scale.

Figure 7: the model input indicates that "transformed allele frequency" was input into the model – what transformation was applied and why?

The transformation was only mentioned in the Supplementary Information (now Appendix 2). We have added the following to the Materials and methods section:

“The observed allele frequency data were transformed through an empirical logit to facilitate approximation by the Gaussian likelihood.”